# GRAPH NEURAL NETWORKS ARE MORE POWERFUL THAN WE THINK

## ABSTRACT

Graph Neural Networks (GNNs) are powerful convolutional architectures that have shown remarkable performance in various node-level and graph-level tasks. Despite their success, the common belief is that the expressive power of standard GNNs is limited and that they are at most as discriminative as the Weisfeiler-Lehman (WL) algorithm. In this paper we argue the opposite and show that standard GNNs, with anonymous inputs, produce more discriminative representations than the WL algorithm. In this direction, we derive an alternative analysis that employs linear algebraic tools and characterize the representational power of GNNs with respect to the eigenvalue decomposition of the graph operators. We prove that GNNs are able to generate distinctive outputs from white uninformative inputs, for, at least, all graphs that have different eigenvalues. We also show that simple convolutional architectures with white inputs, produce features that count the closed paths in the graph and are provably more expressive than the WL representations. Thorough experimental analysis on graph isomorphism and graph classification datasets corroborates our theoretical results and demonstrates the effectiveness of the proposed approach.

## 1 INTRODUCTION

Graph Neural Networks (GNNs) have emerged in the field of machine learning and artificial intelligence as powerful tools that process network structures and network data. Their convolutional architecture allows them to inherit all the favorable properties of convolutional neural networks (CNNs), while they also exploit the graph structure. Despite their remarkable performance, the success of GNNs is still to be demystified. A lot of research has been conducted to theoretically support the experimental developments, focusing on understanding the functionality of GNNs and analyzing their properties. In particular, permutation invariance-equivariance (Maron et al., 2018), stability to perturbations (Gama et al., 2020) and transferability (Ruiz et al., 2020a; Levie et al., 2021) are properties tantamount to the success of the GNNs.

Lately, the research focus has been shifted towards analyzing the expressive power of GNNs, since their universality depends on their ability to produce different outputs for different graphs. The common belief is that standard anonymous GNNs have limited expressive power (Xu et al., 2019) and that it is upper bounded by the expressive power of the Weisfeiler-Lehman (WL) algorithm (Weisfeiler & Leman, 1968). This induced increased research activity towards building more expressive GNNs by either increasing their complexity, or employ independent graph algorithms to design expressive inputs. In this work we argue the opposite. We prove that standard anonymous graph convolutional structures are able to generate more expressive representations than the WL algorithm. Therefore, resorting to handcrafted features or complex GNNs to break the WL limits is not necessary.

Our work is motivated by the following research problem:

**Problem definition:** *Given a pair of different graphs $\mathcal{G}, \hat{\mathcal{G}}$ and anonymous inputs $\boldsymbol{X}, \hat{\boldsymbol{X}}$; is there a GNN $\phi$ with parameter tensor $\mathcal{H}$ such that $\phi\left(\boldsymbol{X}; \mathcal{G}, \mathcal{H}\right), \phi\left(\hat{\boldsymbol{X}}; \hat{\mathcal{G}}, \mathcal{H}\right)$ are nonisomorphic?*

As anonymous inputs, we define inputs that are identity and structure agnostic, i.e., they cannot distinguish graphs or nodes of the graph before processing. *Why anonymous?* Because if the inputs are discriminative prior to processing, concrete conclusions on the discriminative power of GNNs,

cannot be derived. Analyzing GNNs with powerful input features only indicates whether GNNs will maintain or ignore valuable information, not if they can produce this information. This study does not underestimate the importance of drawing powerful input features, which is crucial for most tasks. However, it underscores the need for an alternative analysis.

This paper gives an affirmative answer to the above research question. Our analysis utilizes spectral decomposition tools to show that the source of the WL test as a limit for the expressive power of GNNs is the use of the all-one input. This is expected, since analyzing the representational capacity of $\phi\left(\boldsymbol{X};\mathcal{G},\mathcal{H}\right)$ by studying $\phi\left(\mathbf{1};\mathcal{G},\mathcal{H}\right)$ cannot lead to definitive conclusions. For this reason we study GNNs with white random inputs and show that they generate discriminative outputs, for at least all graphs with different eigenvalues. In particular, we prove that $\phi\left(\boldsymbol{X};\mathcal{G},\mathcal{H}\right), \phi\left(\boldsymbol{X};\hat{\mathcal{G}},\mathcal{H}\right)$ belong to nonisomorphic distributions, even though the input $\boldsymbol{X}$ is drawn from the same distribution. This implies that standard anonymous GNNs are provably more expressive than the WL algorithm as they produce discriminative representations for graphs that fail the WL test, yet have different eigenvalues. In fact, having different eigenvalues is a very mild condition that is rarely not met in practice.

From a practical viewpoint, using white noise as an input to a GNN may be computationally intractable. We show, however, that there are two alternative architectures that are equivalent to a GNN with white random inputs: (i) A GNN that operates on graph representations without requiring any input. (ii) A GNN in which input features are the number of closed paths each node participates. Note that these features can be viewed as the output of the first GNN layer, i.e., they can be generated from a GNN. These results also imply that $\phi\left(\boldsymbol{X};\mathcal{G},\mathcal{H}\right)$ is more powerful than the WL algorithm even if we restrict out attention to countable inputs $\boldsymbol{X}$. Our numerical results show that our proposed GNNs are better anonymous discriminators in some graph classification problems.

Our contribution is summarized as follows:

(C1) We provide a meaningful definition to characterize the representational power of GNNs and develop spectral decomposition tools to study their expressivity.

(C2) We explain that the WL algorithm is not the real limit on the expressive power of anonymous GNNs, but it is associated with the all-one vector as an input.

(C3) We study standard GNNs with white random inputs and show that they can produce discriminative representations for any pair of graphs with different eigenvalues. This implies that standard anonymous GNNs are provably more expressive than the WL algorithm.

(C4) We prove that standard GNNs with white random inputs can count the number of closed paths of each node, which enables the design of equivalent architectures that circumvent the use of random input features.

(C5) We demonstrate the effectiveness of using GNNs with white random inputs, or the proposed alternatives, vs all-one inputs in graph isomorphism and graph classification datasets.

**Related work:** The first work to study the approximation properties of the GNNs was by (Scarselli et al., 2008a). Along the same lines (Maron et al., 2019b; Keriven & Peyré, 2019) discuss the universality of GNNs for permutation invariant or equivariant functions. Then the scientific attention focused on the ability of GNNs to distinguish between nonisomorphic graphs. The works of (Morris et al., 2019; Xu et al., 2019) place the expressive power of GNNs with respect to that of the WL algorithm and prompted various follow-up works in the area. Specifically, (Abboud et al., 2021; Sato et al., 2021) use random features to increase the separation capabilities of GNNs, whereas (Tahmasebi et al., 2020; You et al., 2021; Bouritsas et al., 2022) compute features related to the subgraph information. (Ishiguro et al., 2020) uses label features in WL settings and (Corso et al., 2020; Beaini et al., 2021) use multiple and directional aggregators, respectively, to increase the GNN expressivity. GNNs that use k-tuple and k-subgraph information have been designed by (Maron et al., 2019a; Murphy et al., 2019; Azizian et al., 2020; Morris et al., 2020; Geerts & Reutter, 2021; Giusti et al., 2022). These works use a tensor framework, and employ more expressive structures compared to simple GNNs. However, they are usually computationally heavier to implement and also prone to overfitting. Moreover, (Balcilar et al., 2021) design convolutions in the spectral domain to produce powerful GNNs, whereas (Loukas, 2019) studies the learning capabilities of a GNN with respect to its width and depth. Finally, (Chen et al., 2019) reveal a connection between the universal approximation and the capacity capabilities of GNNs.

## 2 ON THE EXPRESSIVE POWER OF GNNS

One of the most influential works in GNN expressivity by (Xu et al., 2019), compares the representational capabilities of GNNs with those of the WL algorithm (color refinement algorithm). The claim is that GNNs are at most as powerful as the WL algorithm in distinguishing between different graphs. This is indeed true when the input to the GNN is the constant (all-one) vector.

A question that naturally arises is *'Why limit attention to input features $\boldsymbol{x} = \boldsymbol{1}$?'*. The constant vector might be an obvious choice to study anonymous GNNs, however it represents only a small subset of GNN inputs. As we show in the next session, it is also associated with certain spectral limitations, that prohibit a rigorous examination of the GNN representational power. The need for further analysis with general input signals is therefore clear.

To this end, consider graphs $\mathcal{G}$, $\hat{\mathcal{G}}$ with graph operators $\boldsymbol{S}$, $\hat{\boldsymbol{S}} \in \{0,1\}^{N \times N}$. In this paper we focus on graph adjacencies, but any graph operators can be used instead. We assume that $\boldsymbol{S}$, $\hat{\boldsymbol{S}}$ are both symmetric and thus admit eigenvalue decompositions $\boldsymbol{S} = \boldsymbol{U}\boldsymbol{\Lambda}\boldsymbol{U}^T$, $\hat{\boldsymbol{S}} = \hat{\boldsymbol{U}}\hat{\boldsymbol{\Lambda}}\hat{\boldsymbol{U}}^T$, where $\boldsymbol{U}$, $\hat{\boldsymbol{U}}$ are orthogonal matrices containing the eigenvectors, and $\boldsymbol{\Lambda}$, $\hat{\boldsymbol{\Lambda}}$ are the diagonal matrices of corresponding eigenvalues. $\mathcal{G}$, $\hat{\mathcal{G}}$ are nonisomorphic if and only if there is no permutation matrix $\boldsymbol{\Pi}$ such that $\boldsymbol{S} = \boldsymbol{\Pi}\hat{\boldsymbol{S}}\boldsymbol{\Pi}^T$.

A broad class of nonisomorphic graphs have different eigenvalues. To be more precise, let $\mathcal{S}$, $\hat{\mathcal{S}}$ be the set containing the unique eigenvalues of $\boldsymbol{S}$ and $\hat{\boldsymbol{S}}$ with multiplicities denoted by $m_\lambda$, $\hat{m}_\lambda$ respectively. The following assumption is heavily used in the main part of this paper:

**Assumption 2.1** $\boldsymbol{S}$, $\hat{\boldsymbol{S}}$ *have different eigenvalues, i.e., there exists $\lambda \in \mathcal{S}$, such that $\lambda \notin \hat{\mathcal{S}}$ or $m_\lambda \neq \hat{m}_\lambda$.*

When Assumption 2.1 holds, $\mathcal{G}$, $\hat{\mathcal{G}}$ are always nonisomorphic. Assumption 2.1 is not restrictive. Real nonisomorphic graphs have different eigenvalues with very high probability (Haemers & Spence, 2004). Corner cases where Assumption 2.1 doesn't hold are studied in Appendix H. First, we consider GNNs that are constructed by the following modules, corresponding to the neurons of a typical (non-graph) neural network:

$$\boldsymbol{Y} = \sigma \left( \sum_{k=0}^{K-1} \boldsymbol{S}^k \boldsymbol{X} \boldsymbol{H}_k \right). \tag{1}$$

The module in (1) is composed by a graph filter of length $K$ followed by a nonlinearity $\sigma(\cdot)$. $\boldsymbol{H}_k$ represents the filter parameters and can be a matrix, a vector, or a scalar. In order to characterize the representational power of GNNs with general input $\boldsymbol{X} \in \mathbb{R}^{N \times D}$, we provide the following theorem:

**Theorem 2.2** *Let $\mathcal{G}$, $\hat{\mathcal{G}}$ be nonisomorphic graphs with graph signals $\boldsymbol{X}$, $\hat{\boldsymbol{X}}$. Also let $\boldsymbol{V}_\lambda$, $\hat{\boldsymbol{V}}_\lambda$ be the eigenspaces corresponding to $\lambda$ in $\boldsymbol{S}$, $\hat{\boldsymbol{S}}$ respectively. There exist a GNN $\phi(\boldsymbol{X}; \mathcal{G}, \mathcal{H})$ that produces nonisomorphic representations for $\mathcal{G}$ and $\hat{\mathcal{G}}$ if:*

1. *There does not exist permutation matrix $\boldsymbol{\Pi}$ such that $\boldsymbol{X} = \boldsymbol{\Pi}\hat{\boldsymbol{X}}$, or*

2. *There exists $\lambda \in \mathcal{S}$, such that $\lambda \notin \hat{\mathcal{S}}$ and $\boldsymbol{X}^T \boldsymbol{V}_\lambda \neq \boldsymbol{0}$, or*

3. *There exists $\lambda \in \mathcal{S}, \hat{\mathcal{S}}$, such that $m_\lambda \neq \hat{m}_\lambda$ and $\boldsymbol{X}^T \left( \boldsymbol{V}_\lambda \oplus \hat{\boldsymbol{V}}_\lambda \right) \neq \boldsymbol{0}$.[1]*

Theorem 2.2 highlights the importance of the input $\boldsymbol{X}$ in the representational capabilities of a GNN. For problems in which inputs are given, it states that a GNN can distinguish between nonisomorphic graphs if they have different graph signals or their signals are not orthogonal to the eigenspace associated with the eigenvalue that differentiates them. In problems where inputs are not available, Theorem 2.2 provides guidelines on how to design input $\boldsymbol{X}$ from the graph.

Theorem 2.2 also indicates that the limitations of GNNs discussed in (Xu et al., 2019) are not due to the architecture but they are limitations associated with the input. In particular, $\boldsymbol{x} = \boldsymbol{1}$ fails to satisfy condition 1, while it is also prone to fail condition 2 and 3, since the majority of real graphs have eigenvectors that are orthogonal to $\boldsymbol{1}$. This observation is the impetus to study GNNs with white

---

[1]We define $\boldsymbol{V}_\lambda \oplus \hat{\boldsymbol{V}}_\lambda := \{\boldsymbol{u} + \boldsymbol{w} \mid \boldsymbol{u} \in \boldsymbol{V}_\lambda; \boldsymbol{w} \in \hat{\boldsymbol{V}}_\lambda; \boldsymbol{u}, \boldsymbol{w} \notin \boldsymbol{V}_\lambda \bigcap \hat{\boldsymbol{V}}_\lambda\}$ as the exclusive sum of subspaces

random inputs. White inputs are unonymous (they carry no information on the graphs), they model a large set of GNN inputs and always satisfy the conditions of Theorem 2.2. Furthermore, as we show in section 5, GNNs with random inputs can generate deterministic, countable features:

$$\boldsymbol{X} = \left[ \text{diag}\left(\boldsymbol{S}^0\right), \text{diag}\left(\boldsymbol{S}^1\right), \text{diag}\left(\boldsymbol{S}^2\right), \ldots, \text{diag}\left(\boldsymbol{S}^{D-1}\right)\right] \in \mathbb{N}_0^{N \times D}, \tag{2}$$

that satisfy the conditions of Theorem 2.2. A nice interpretation of this result, given in section 5, connects $\boldsymbol{X}$ in (2) with high-order subgraphs and shows that GNNs can count closed paths.

## 3    LIMITATIONS OF GNNS WITH $\boldsymbol{x} = \boldsymbol{1}$ INPUT AND THE WL ALGORITHM

Using Theorem 2.2 we can explain why feeding a GNN with $\boldsymbol{x} = \boldsymbol{1}$ is limiting. The limitations associated with input $\boldsymbol{x} = \boldsymbol{1}$ are also highly related to the limitations of the WL algorithm. The problem appears in graphs that admit spectral decompositions with eigenvectors that are orthogonal to $\boldsymbol{1}$ (they sum up to zero). According to Theorem 2.2, if two graphs are the same except eigenvalues corresponding to eigenvectors that sum up to zero, GNNs with constant inputs are likely to produce isomorphic representations for the two graphs. To see this consider the graphs $\mathcal{G}$, $\hat{\mathcal{G}}$ with spectral decompositions:

$$\boldsymbol{S} = \boldsymbol{U}\boldsymbol{\Lambda}\boldsymbol{U}^T = \lambda_1\boldsymbol{u}_1\boldsymbol{u}_1^T + \lambda_2\boldsymbol{u}_2\boldsymbol{u}_2^T + \lambda_3\boldsymbol{u}_3\boldsymbol{u}_3^T, \tag{3}$$

$$\hat{\boldsymbol{S}} = \hat{\boldsymbol{U}}\hat{\boldsymbol{\Lambda}}\hat{\boldsymbol{U}}^T = \lambda_1\boldsymbol{u}_1\boldsymbol{u}_1^T + \lambda_2\boldsymbol{u}_2\boldsymbol{u}_2^T + \hat{\lambda}_3\boldsymbol{u}_3\boldsymbol{u}_3^T, \tag{4}$$

where $\lambda_3 \neq \hat{\lambda}_3$. If $\boldsymbol{u}_3$ is orthogonal to $\boldsymbol{1}$ then:

$$\boldsymbol{S}^k\boldsymbol{1} = \boldsymbol{U}\boldsymbol{\Lambda}^k\boldsymbol{U}^T\boldsymbol{1} = \lambda_1^k\boldsymbol{u}_1\boldsymbol{u}_1^T\boldsymbol{1} + \lambda_2^k\boldsymbol{u}_2\boldsymbol{u}_2^T\boldsymbol{1} + \lambda_3^k\boldsymbol{u}_3\boldsymbol{u}_3^T\boldsymbol{1} = \lambda_1^k\left(\boldsymbol{u}_1^T\boldsymbol{1}\right)\boldsymbol{u}_1 + \lambda_2^k\left(\boldsymbol{u}_2^T\boldsymbol{1}\right)\boldsymbol{u}_2 \tag{5}$$

$$\hat{\boldsymbol{S}}^k\boldsymbol{1} = \hat{\boldsymbol{U}}\hat{\boldsymbol{\Lambda}}^k\hat{\boldsymbol{U}}^T\boldsymbol{1} = \lambda_1^k\boldsymbol{u}_1\boldsymbol{u}_1^T\boldsymbol{1} + \lambda_2^k\boldsymbol{u}_2\boldsymbol{u}_2^T\boldsymbol{1} + \hat{\lambda}_3^k\boldsymbol{u}_3\boldsymbol{u}_3^T\boldsymbol{1} = \lambda_1^k\left(\boldsymbol{u}_1^T\boldsymbol{1}\right)\boldsymbol{u}_1 + \lambda_2^k\left(\boldsymbol{u}_2^T\boldsymbol{1}\right)\boldsymbol{u}_2 \tag{6}$$

The diffused information in GNNs with this naive input is related to $\boldsymbol{S}^k\boldsymbol{1}$ and therefore in the above example the decisive information that differentiates the two graphs is highly likely to be omitted.

Graphs with eigenvectors orthogonal to $\boldsymbol{1}$ can also affect the performance of the WL algorithm. In the absence of features the WL algorithm is initialized with $\boldsymbol{x} = \boldsymbol{S}\boldsymbol{1}$, which is propagated through the nodes iteratively. In graphs with eigenvectors orthogonal to $\boldsymbol{1}$, the propagated degrees have suffered critical information loss in the initialization, which in certain graph structures is impossible to recover, as WL iterations progress. Further analysis on this subject can be found in Appendix C.

Classic examples of graphs with different eigenvalues, that the WL algorithm and GNNs with $\boldsymbol{x} = \boldsymbol{1}$ input cannot tell apart, are presented in Figs. 1, 2. In particular, these approaches decide that $\mathcal{G}$ and $\hat{\mathcal{G}}$ in Fig. 1 and $\mathcal{G}$ and $\hat{\mathcal{G}}$ in Fig. 2 are the same. This is due to the fact that these graphs contain eigenvectors that are orthogonal to $\boldsymbol{1}$. The case of Fig. 1 is straightforward. All

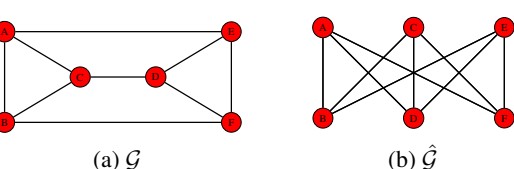

(a) $\mathcal{G}$               (b) $\hat{\mathcal{G}}$

Figure 1: WL indistinguishable graphs.

the nodes of $\mathcal{G}$ and $\hat{\mathcal{G}}$ have the same degree, i.e., $\boldsymbol{x} = \boldsymbol{1}$ is an eigenvector in both graphs and thus orthogonal to all the remaining eigenvectors. As a result, the node degrees (which are the same for both graphs) are the only information that the WL algorithm and GNNs with $\boldsymbol{1}$ input are able to process. The case of Fig. 2 is more complicated; $\boldsymbol{x} = \boldsymbol{1}$ is not an eigenvector in any of the graphs, but it is orthogonal to the eigenvectors corresponding to the eigenvalues that differentiate the two graphs. Consequently, the operation $\boldsymbol{S}\boldsymbol{1}$ negates vital information and the two approaches fail.

Detailed information about the eigenvalues and eigenvectors of the graphs in Figs. 1, 2 can be found in Tables 8, 9 of Appendix K. This information corroborates the issues discussed in the previous paragraph. As noted earlier and will be explained in more detail in the upcoming sections, GNNs are discriminative enough to overcome these issues and provide nonisomorphic representation for $\mathcal{G}$ and $\hat{\mathcal{G}}$ in both Figs. 1, 2.

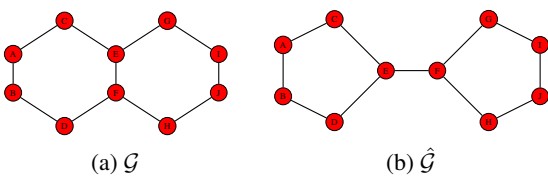

(a) $\mathcal{G}$               (b) $\hat{\mathcal{G}}$

Figure 2: WL indistinguishable graphs

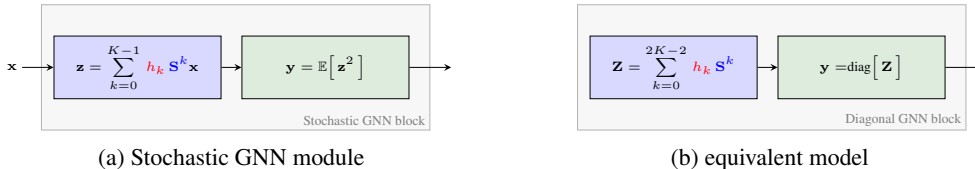

(a) Stochastic GNN module        (b) equivalent model

Figure 3: GNN with random Gaussian input

## 4   FEEDING THE GNN WITH RANDOM INPUT

In this section we study the representational power of GNNs, by feeding them with random white inputs. Our analysis overcomes the GNN limitations associated with $\boldsymbol{x} = \boldsymbol{1}$ and derives rigorous conclusions. Consider the GNN module in (1) where $\boldsymbol{H}_k$ is a scalar, i.e., $\boldsymbol{y} = \sigma \left( \sum_{k=0}^{K-1} h_k \boldsymbol{S}^k \boldsymbol{x} \right)$. Before choosing an appropriate nonlinearity, let us focus on the linear convolutional graph filter of length $K$:

$$\boldsymbol{z} = \sum_{k=0}^{K-1} h_k \boldsymbol{S}^k \boldsymbol{x}, \tag{7}$$

which we load with white random input $\boldsymbol{x} \in \mathbb{R}^N$, i.e., $\mathbb{E}[\boldsymbol{x}] = 0$, $\mathbb{E}[\boldsymbol{x}\boldsymbol{x}^T] = \sigma^2 \boldsymbol{I}$. Since $\boldsymbol{x}$ is a zero-mean random vector, $\boldsymbol{z}$ is also a random vector with $\mathbb{E}[\boldsymbol{z}] = \boldsymbol{0}$. Thus, the expected value provides no information about the network. Measuring the covariance, on the other hand, yields:

$$\text{cov}[\boldsymbol{z}] = \mathbb{E}\left[\boldsymbol{z}\boldsymbol{z}^T\right] = \mathbb{E}\left[\sum_{k=0}^{K-1} h_k \boldsymbol{S}^k \boldsymbol{x}\boldsymbol{x}^T \sum_{m=0}^{K-1} h_m \boldsymbol{S}^{m^T}\right] = \sum_{k=0}^{K-1} h_k \boldsymbol{S}^k \mathbb{E}\left[\boldsymbol{x}\boldsymbol{x}^T\right] \sum_{m=0}^{K-1} h_m \boldsymbol{S}^m$$

$$= \sigma^2 \sum_{k=0}^{K-1} h_k \boldsymbol{S}^k \sum_{m=0}^{K-1} h_m \boldsymbol{S}^m = \sigma^2 \sum_{k=0}^{K-1}\sum_{m=0}^{K-1} h_k h_m \boldsymbol{S}^k \boldsymbol{S}^m = \sum_{k=0}^{2K-2} h_k' \boldsymbol{S}^k, \tag{8}$$

where $h_k' = \sigma^2 \sum_{m,l} h_m h_l$, such that $m + l = k$. The results of equation (8) are noteworthy. We have shown that the covariance of a graph filter with random white input corresponds to a different graph filter with no input. Furthermore, the resulting filter has length $2K - 1$, whereas the original filter has length $K$. In other words the nonlinearity introduced by the covariance computation enables the filter to gather information from a broader neighborhood compared to the initial filter. However, there is a caveat that the degrees of freedom for $h'$ are $K$ and not $2K - 1$. Further discussion on the subject can be found in Appendix D.

In practice we want to associate the output of a GNN with a feature for each node that is permutation equivariant. This is not the case with the rows or columns of the covariance matrix in (8). Therefore we choose $\sigma(\cdot)$ to be the variance of each node i.e.,

$$\boldsymbol{y} = \sigma(\boldsymbol{z}) = \text{var}[\boldsymbol{z}] = \mathbb{E}\left[\boldsymbol{z}^2\right] = \text{diag}\left(\text{cov}[\boldsymbol{z}]\right) = \text{diag}\left(\sum_{k=0}^{2K-2} h_k' \boldsymbol{S}^k\right) = \sum_{k=0}^{2K-2} h_k' \text{diag}\left(\boldsymbol{S}^k\right). \tag{9}$$

The stochastic GNN module, defined by the linear filter in (7) and the variance operator is illustrated in Fig. 3a. Regarding its expressive power, we present the following theorem:

**Theorem 4.1** *Let $\mathcal{G}$, $\hat{\mathcal{G}}$ be nonisomorphic graphs. If Assumption 2.1 holds, there exists a GNN with modules as in Fig. 3a that produces nonisomorphic representations for the two graphs.*

The implications of Theorem 4.1 are noteworthy. A GNN $\phi(\boldsymbol{X}; \mathcal{G}, \mathcal{H})$ with white input produces outputs that are drawn from different distributions for all graphs with different eigenvalues. Furthermore, measuring the variance produces equivariant node representations that can separate all graphs with different eigenvalues.

**Proposition 4.1** *The GNN module in Fig. 3a with white random input is equivalent to the GNN module in Fig. 3b with no input up to degrees of freedom (dependencies) in the filter parameters.*

The proof of Proposition 4.1 is by the definition (equation (9)) of the GNN module in Fig. 3b. The claim is eminent. It proves equivalence of two GNN architectures; a standard graph filter with white input followed by a variance operator with a deterministic graph filter followed by a diagonal operator. Depending on the problem and the variance of the system one has the option to choose either of them. Further discussion on the stochastic approach can be found in Appendix D.

## 5 THE DIAGONAL MODULE

Proposition 4.1 proved the equivalence of the two GNN modules in Fig. 3. In this section we focus on the module in 3b and analyze its unique properties. To be more precise, we study the following diagonal GNN module:

$$\boldsymbol{y} = \sigma \left( \sum_{k=0}^{K-1} h_k \text{diag} \left( \boldsymbol{S}^k \right) \right), \tag{10}$$

Note that the module in (10) is not exactly the same as the one in Fig. 3b, since a nonlineatity is added and the filter is of length $K$. As an example, we test the proposed diagonal module on the graphs of Figs. 1, 2, and present the output $\boldsymbol{y}$ of (10) with parameters $(h_0, h_1, h_2, h_3, h_4, h_5) = \left(10, 1, -\frac{1}{2}, \frac{1}{3}, -\frac{1}{4}, \frac{1}{5}\right)$ and ReLU nonlinearity, in Table 1.

Table 1: Outputs $\boldsymbol{y}$ of $\mathcal{G}$ and $\hat{\boldsymbol{y}}$ of $\hat{\mathcal{G}}$ of the proposed diagonal module for the graphs in Figs. 1, 2.

| GRAPH | | NODE | | | | | | | | | |
|---|---|---|---|---|---|---|---|---|---|---|---|
| | | A | B | C | D | E | F | G | H | I | J |
| FIG. 1 | $\boldsymbol{y}$ | 10.42 | 10.42 | 10.42 | 10.42 | 10.42 | 10.42 | - | - | - | - |
| | $\hat{\boldsymbol{y}}$ | 1.75 | 1.75 | 1.75 | 1.75 | 1.75 | 1.75 | - | - | - | - |
| FIG. 2 | $\boldsymbol{y}$ | 7.5 | 7.5 | 7.25 | 7.25 | 5.25 | 5.25 | 7.25 | 7.25 | 7.5 | 7.5 |
| | $\hat{\boldsymbol{y}}$ | 7.9 | 7.9 | 7.65 | 7.65 | 5.65 | 5.65 | 7.65 | 7.65 | 7.9 | 7.9 |

We observe that the output (10) of the proposed diagonal module produces embeddings that are different for the nodes of $\mathcal{G}$ and $\hat{\mathcal{G}}$ in both Figs. 1, 2. Therefore, there does not exist permutation matrix $\boldsymbol{\Pi}$ such that $\boldsymbol{y} = \boldsymbol{\Pi}\hat{\boldsymbol{y}}$ and the proposed architecture is able to tell $\mathcal{G}$ and $\hat{\mathcal{G}}$ apart in both Figs. 1, 2. This is in stark contrast to GNNs with $\boldsymbol{x} = \boldsymbol{1}$ input and the WL algorithm that fail to distinguish between these graphs (as discussed in section 3). We now study the diagonal module in the frequency domain to analyse the representational capabilities of standard GNNs:

$$\boldsymbol{y} = \sigma \left( \sum_{k=0}^{K-1} h_k \text{diag} \left( \sum_{n=1}^{N} \lambda_n{}^k \boldsymbol{u}_n \boldsymbol{u}_n^T \right) \right) = \sigma \left( \sum_{k=0}^{K-1} \sum_{n=1}^{N} h_k \lambda_n^k |\boldsymbol{u}_n|^2 \right) = \sigma \left( \sum_{n=1}^{N} \tilde{h} \left( \lambda_n \right) |\boldsymbol{u}_n|^2 \right), \tag{11}$$

where $\tilde{h} \left( \lambda_n \right) = \sum_{k=0}^{K-1} h_k \lambda_n^k$ is the frequency response of the graph filter in (7) at $\lambda_n$. In simple words, the frequency representation of the proposed diagonal module, or standard GNNs with white input, depends on the absolute values of the graph adjacency eigenvectors. On the contrary, standard GNNs with constant inputs admit a different frequency representation:

$$\boldsymbol{y}_1 = \sigma \left( \sum_{k=0}^{K-1} h_k \boldsymbol{S}^k \boldsymbol{1} \right) = \sigma \left( \sum_{k=0}^{K-1} \sum_{n=1}^{N} h_k \lambda_n{}^k \boldsymbol{u}_n \boldsymbol{u}_n^T \boldsymbol{1} \right) = \sigma \left( \sum_{n=1}^{N} \tilde{h} \left( \lambda_n \right) \boldsymbol{u}_n^T \boldsymbol{1} \boldsymbol{u}_n \right), \tag{12}$$

As we can see both outputs $\boldsymbol{y}$, $\boldsymbol{y}_1$ are functions of the graph eigenvectors. The question that arises is which function, $|\boldsymbol{u}_n|$ or $\left( \boldsymbol{u}_n^T \boldsymbol{1} \right) \boldsymbol{u}_n$, results in more expressive GNNs. The naive answer is that depending on the graph, there is a trade-off between the information loss caused by $|\boldsymbol{u}_n|$ or $\left( \boldsymbol{u}_n^T \boldsymbol{1} \right) \boldsymbol{u}_n$. However, after adding a second layer, GNNs with white inputs are always more powerful than GNNs initialized by $\boldsymbol{1}$. This will be explained in more detail in the next section.

A closer look at equations (10) and (11), reveals further insights regarding standard GNNs with anonymous inputs. In particular,

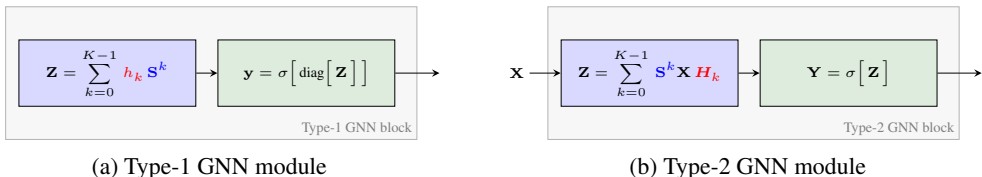

(a) Type-1 GNN module           (b) Type-2 GNN module

Figure 4: Proposed GNN modules

**Theorem 5.1** *A standard GNN $\phi\left(\boldsymbol{X};\mathcal{G},\mathcal{H}\right)$ as defined in* (1) *can count all the closed paths each node participates, if initialized with white input.*

The proof is the combination of Theorem 4.1 and equation (10). In particular, a standard anonymous GNN can compute the following vector representations:

$$\boldsymbol{d}^k = \text{diag}\left(\boldsymbol{S}^k\right) = \sum_{n=1}^{N} \lambda_n^k |\boldsymbol{u}_n|^2, \tag{13}$$

that count the number of $k-$ length closed paths of each node. For instance, when $k = 2$, $\boldsymbol{d}^k$ indicates the degree of each node, whereas for $k = 3$, it counts the number of triangles each node is involved in, multiplied by a constant factor. For $k = 4$, $\boldsymbol{d}^k$ holds information about the degrees of $1-$hop and $2-$hop neighbors as well as the $4-$th order cycles. Similar observations are derived by considering larger values of $k$. Graph adjacency diagonals are not only associated with $k-$hop neighbor degrees but also with motifs that are present in the graph. This observation becomes even more valuable, if we consider the significance of subgraph mining in graph theory (Kuramochi & Karypis, 2001; Danisch et al., 2018). Our final observation is that, the $k-$th order closed paths are associated with the absolute values of the adjacency eigenvectors $|\boldsymbol{u}_n|$, whereas degrees are connected with $\left(\boldsymbol{u}_n^T \mathbf{1}\right) \boldsymbol{u}_n$.

The following theorem characterizes the expressive power of GNNs with modules as in (10):

**Theorem 5.2** *Let $\mathcal{G}$, $\hat{\mathcal{G}}$ be nonisomorphic graphs. If Assumption 2.1 holds, there exists a GNN with diagonal modules as in* (10) *that produces distinct representations for $\mathcal{G}$, $\hat{\mathcal{G}}$.*

## 6   DESIGNING POWERFUL GNN ARCHITECTURES

After analyzing GNNs with white inputs and introducing the GNN module in (10), it is time to build practical powerful architectures. The modules we employ to build the proposed GNN architecture are presented in Fig. 4. Regarding their functionality we provide the following result:

**Proposition 6.1** *A GNN designed with the diagonal modules of Fig. 4a (eq.* (10)*) in the input layer is equivalent to a standard GNN designed with the modules of Fig. 4b in the input layer, if the input to the modules of Fig. 4b (eq.* (1)*) is designed according to:*

$$\boldsymbol{X} = \left[ diag\left(\boldsymbol{S}^0\right), diag\left(\boldsymbol{S}^1\right), diag\left(\boldsymbol{S}^2\right), \ldots, diag\left(\boldsymbol{S}^{D-1}\right) \right]. \tag{14}$$

The claim of Proposition 6.1 is fundamental and relates a standard GNN with white input to a standard GNN with countable input defined by (31). Specifically, combining propositions 4.1 and 6.1 yields a direct connection between the three considered architectures; standard GNNs with white input and variance nonlinearity, GNNs with no input and diagonal operator, and standard GNNs with input as in (14). Guided by these findings we design the GNN architectures presented in Fig. 5. The architecture on the left uses one type of GNN blocks (type-2) and the input is designed by equation (14). Furthermore, it is a symmetric architecture and admits all the favorable properties of symmetric designs. On the other hand, the architecture on the right uses a combination of type-1 and type-2 GNN blocks and designing an input is not necessary. Although the design is not symmetric, it offers reduced number of trainable parameters and reuse of first layer features, which has been observed to benefit convolutional architectures. The expressive power of the proposed architectures is demonstrated in the following theorem:

**Theorem 6.1** *Let $\mathcal{G}$, $\hat{\mathcal{G}}$ be nonisomorphic graphs with graph signals $\boldsymbol{X}$, $\hat{\boldsymbol{X}}$ designed according to* (14). *If Assumption 2.1 holds, then the proposed GNNs in Fig. 5 can tell the two graphs apart.*

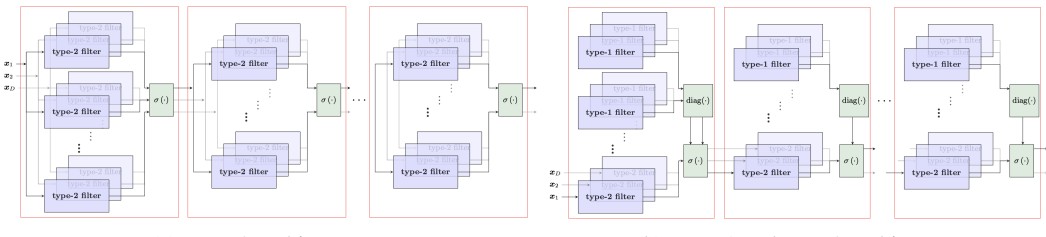

| (a) Type-2 architecture | (b) Type-1 and type-2 architecture |

Figure 5: Proposed GNN architectures

**Corollary 6.2** *The proposed architectures in Fig. 5 are strictly more expressive compared to GNNs with $x = 1$ or $x = S1$ inputs.*

Corollary 6.2 follows from Theorem 6.1 and the fact that both $\mathrm{diag}\left(S^0\right) = 1$, $\mathrm{diag}\left(S^2\right) = S1$ are included in the proposed input $X$, defined in (14).

Overall, our proposed analysis proves that standard GNNs $\phi\left(X; \mathcal{G}, \mathcal{H}\right)$ are more powerful than the WL algorithm for both countable and continuous inputs.

## 7 SIMULATIONS

In this section we test the effect of using anonymous all-one inputs vs anonymous random inputs on the expressivity of GNNs. The task of interest is graph classification. In particular, we use graph isomorphism and graph classification datasets and train the standard convolutional GNN in (1) and GIN (Xu et al., 2019). GIN initialized with $x = 1$ is denoted as $\text{GIN}_1$ and GIN with random input is denoted as $\text{GIN}_{\text{plus}}$. For the standard GNN model we only test random inputs. To avoid practical issues associated with random inputs we use the equivalent model of section 6 instead, i.e., we intialize both standard convolutional GNN and GIN according to equation (14).

### 7.1 THE CSL DATASET

Our first experiment involves the Circular Skip Link (CSL) dataset, which was introduced in (Murphy et al., 2019) to test the expressiveness of GNNs; it is the golden standard when it comes to benchmarking GNNs for isomorphism (Dwivedi et al., 2020). CSL is a symmetric graph dataset. It contains 150 4-regular graphs, where the edges form a cycle and contain skip-links between nodes. A schematic representation of the CSL graphs can be found in Appendix K. Each graph consists of 41 nodes and 164 edges and belongs to one of 10 classes. All the nodes have degree 4 and thus $x = 1$ is an eigenvector of every graph and orthogonal to all the remaining eigenvectors. As a result the degree vector is uninformative and so is any message passing operation of the degree.

GNNs initialized with $x = 1$ and the WL algorithm fail to provide any essential information for this set of graphs and the classification task is completely random, as shown in Table 4. The proposed GNN architectures, on the other hand, have no issue in dealing with this dataset. In particular a single diagonal GNN module with parameters $(h_0, h_1, h_2, h_3, h_4, h_5, h_6, h_7, h_8, h_9) = \left(0, 1, -\frac{1}{2}, \frac{1}{3}, -\frac{1}{4}, \frac{1}{5}, -\frac{1}{6}, \frac{1}{7}, -\frac{1}{8}, \frac{1}{9}\right)$ and $\sigma(\cdot)$ being the linear function, is able to classify these graphs with $100\%$ accuracy. To see this, we present in Table 2 the output $1^T y$ for every class, where $y$ is defined in (10) with the aforementioned parameters. The output is the same for each graph in the same class but different for graphs that belong to different classes. Therefore, perfect classification accuracy is achieved by passing the GNN output to a simple linear classifier or even a linear assignment algorithm.

Table 2: GNN output $y$ for every class of the CSL graphs.

| | | | | | CLASS | | | | |
|---|---|---|---|---|---|---|---|---|---|
| 0 | 1 | 2 | 3 | 4 | 5 | 6 | 7 | 8 | 9 |
| 73616 | -45968 | 1059 | -30593 | -25345 | -26001 | -17555 | -28543 | 16065 | -21163 |

## 7.2 Social and biological networks

Next, we test the performance of the proposed architecture with standard social, chemical and bioinformatics graph classification datasets (Errica et al., 2019). The details of each dataset can be found in Table 3. To perform the graph classification task, we train a GNN with 4 layers, each layer consisting of the same number of neurons. The input to each GNN is designed by equation (14) with $K = 10$ and we also pass the $k-$th degree vector. Apart from feeding the output of each layer to the next layer, we also apply a readout function that performs graph pooling. The graph pooling layer generates a global graph embedding from the node representations and passes it to a linear classifier. The nonlinearity is chosen to be the ReLU. An illustration of the used architecture, as well as a detailed description of the experiments, is presented in Appendix K.

To test the performance of the anonymous architectures we divide each dataset into $50 - 50$ training-testing splits and perform 10-fold cross validation. We measure the micro F1 and macro F1 score for each epoch and present the epoch with the best average result among the 10 folds. The mean and standard deviation of the testing results over 10 shuffles are presented in Table 4. In Table 4 we ob-

Table 3: Datasets

| Dataset | # Graphs | Average # Vertices | Average # Edges | # Classes | Network Type |
|---|---|---|---|---|---|
| CSL | 150 | 41 | 164 | 10 | Circulant |
| IMDBBINARY | 1,000 | 20 | 193 | 2 | Social |
| IMDBMULTI | 1,500 | 13 | 132 | 3 | Social |
| REDDITBINNARY | 2000 | 430 | 498 | 2 | Social |
| REDDITMULTI | 5000 | 509 | 595 | 5 | Social |
| PTC | 344 | 26 | 52 | 3 | Bioinformatic |
| PROTEINS | 1,113 | 39 | 146 | 2 | Bioinformatic |
| MUTAG | 188 | 18 | 20 | 2 | Chemical |
| NCI1 | 4110 | 39 | 73 | 2 | Chemical |

Table 4: Average testing score and standard deviation over 10 shuffles

| Dataset | Proposed micro F1 | Proposed macro F1 | GIN micro F1 | GIN macro F1 | $\text{GIN}_{\text{plus}}$ (proposed+GIN) micro F1 | $\text{GIN}_{\text{plus}}$ (proposed+GIN) macro F1 |
|---|---|---|---|---|---|---|
| CSL | $\mathbf{100 \pm 0}$ | $\mathbf{100 \pm 0}$ | $10 \pm 3.3$ | $1.8 \pm 0.6$ | $\mathbf{100 \pm 0}$ | $\mathbf{100 \pm 0}$ |
| IMDBBINARY | $71.7 \pm 2.5$ | $71.3 \pm 2.7$ | $\mathbf{74.7 \pm 3.2}$ | $\mathbf{74.6 \pm 3.2}$ | $71.6 \pm 3.4$ | $\mathbf{71 \pm 3.8}$ |
| IMDBMULTI | $46.1 \pm 2.8$ | $44.2 \pm 3.2$ | $\mathbf{50.3 \pm 2.8}$ | $\mathbf{48 \pm 3.4}$ | $48.6 \pm 2.9$ | $46.1 \pm 4.2$ |
| REDDITBINARY | $87.2 \pm 4.1$ | $87.1 \pm 4.3$ | $81.6 \pm 5.6$ | $81.5 \pm 5.7$ | $\mathbf{89.8 \pm 2.3}$ | $\mathbf{89.7 \pm 2.3}$ |
| REDDITMULTI | $54 \pm 2.2$ | $52.4 \pm 2.1$ | $52.4 \pm 2.4$ | $50.9 \pm 2.4$ | $\mathbf{55 \pm 1.5}$ | $\mathbf{53.6 \pm 1.7}$ |
| PTC | $63.6 \pm 4.9$ | $61.4 \pm 6.9$ | $\mathbf{65.7 \pm 8.8}$ | $\mathbf{65.1 \pm 9.1}$ | $62.5 \pm 5.1$ | $61.4 \pm 5.5$ |
| PROTEINS | $74.2 \pm 4.2$ | $73 \pm 4$ | $74 \pm 4.6$ | $72.3 \pm 4.5$ | $\mathbf{74.3 \pm 4.8}$ | $\mathbf{73.1 \pm 4.5}$ |
| MUTAG | $89.3 \pm 7.3$ | $87.2 \pm 9.3$ | $\mathbf{89.8 \pm 7.6}$ | $88.6 \pm 8.8$ | $\mathbf{89.8 \pm 8}$ | $\mathbf{88.7 \pm 8.6}$ |
| NCI1 | $74.5 \pm 2.1$ | $74.3 \pm 2.1$ | $\mathbf{77.2 \pm 1.9}$ | $\mathbf{77.2 \pm 1.9}$ | $76.3 \pm 3.7$ | $76.2 \pm 3.8$ |

serve that the proposed architecture and $\text{GIN}_{\text{plus}}$ markedly outperform $\text{GIN}_1$ in the REDDITBINARY dataset, and also show notable improvement in the REDDITMULTI dataset. $\text{GIN}_1$, on the other hand, has a $3\%$ advantage in the IMDBBINARY dataset, whereas in the remaining datasets the performances of the competing algorithms are statistically similar. The latter can be explained, since the vital classification components, of these datasets, are not orthogonal to $x = 1$ and $\text{GIN}_1$ is not undergoing critical information loss. Overall, we conclude that properly designed GNNs, as the proposed and $\text{GIN}_{\text{plus}}$ can not only demonstrate remarkable performance in graph classification tasks, but can also handle pathological datasets such as the CSL. This is an indicator on the importance of the representational properties. However, what is equally important is generalization capability, data handling and optimization, which we do not study in this paper.

## 8 Conclusion

In this paper we studied the expressive power of GNNs with spectral decomposition tools. We showed that, contrary to common belief, the WL algorithm is not the real limit and proved that anonymous GNNs can distinguish between any graphs with different eigenvalues. Furthermore, we explained the limitations of GNNs with all-one inputs and designed GNN architectures that overcome these limitations. Experiments with graph isomorphism and graph classification datasets demonstrated the validity of the proposed approach. With this work we move one step closer to understanding the properties of GNNs and analyzing their functionality.

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

# Appendix

## Table of Contents

## A PRELIMINARIES

Networks are naturally represented by graphs $\mathcal{G} := (\mathcal{V}, \mathcal{E})$, where $\mathcal{V} = \{1, \ldots, N\}$ is the set of vertices (nodes) and $\mathcal{E} = \{(v, u)\}$ are the edges between pairs of nodes. The 1-hop neighborhood $\mathcal{N}(v)$ of node $v$ is the set of nodes $u \in \mathcal{V}$ that satisfy $(u, v) \in \mathcal{E}$. A graph can also be modeled by a Graph Shift Operator (GSO) $\boldsymbol{S} \in \mathbb{R}^{N \times N}$, where $\boldsymbol{S}(i, j)$ quantifies the relation between node $i$ and node $j$ and $N = |\mathcal{V}|$. Popular choices of the GSO is the graph adjacency, the graph Laplacian or weighted versions of them. The nodes of the graph are often associated with graphs signals $\boldsymbol{X} \in \mathbb{R}^{N \times D}$, also known as node attributes, where $D$ is the dimension of each graph signal (feature dimension).

## A.1 GRAPH NEURAL NETWORKS (GNNS)

A graph convolution is defined as:

$$z = \sum_{k=0}^{K-1} h_k S^k x,\tag{15}$$

where $H(S) = \sum_{k=0}^{K-1} h_k S^k$ is a linear filter of length $K$ and $x$, $z \in \mathbb{R}^N$ are the input and output of the filter respectively. Let $S = U \Lambda U^T$, be the eigenvalue decomposition of $S$. Then:

$$z = \sum_{k=0}^{K-1} h_k U \Lambda^k U^T x\tag{16}$$

$$U^T z = \sum_{k=0}^{K-1} h_k \Lambda^k U^T x\tag{17}$$

$$\tilde{z} = \sum_{k=0}^{K-1} h_k \Lambda^k \tilde{x},\tag{18}$$

where $\tilde{x}$, $\tilde{z}$ are the frequency representations of $x$, $z$ respectively. The frequency representation of the graph filter is $\tilde{H}(\Lambda) = \sum_{k=0}^{K-1} h_k \Lambda^k$ and can also be written as:

$$\tilde{H}(\lambda_i) = \sum_{k=0}^{K-1} h_k \lambda_i^k.\tag{19}$$

$\tilde{H}(\lambda_i)$ is a polynomial on $\lambda_i$ and $\tilde{z}_i = \tilde{H}(\lambda_i) \tilde{x}_i$. The simplest form of a Graph Neural Network (GNN) is an array of graph filters followed by point-wise nonlinearities. The $l$-th layer of the GNN is a graph perceptron, which is described by:

$$X^{(l+1)} = \sigma \left( \sum_{k=0}^{K-1} h_k^{(l)} S^k X^{(l)} \right).\tag{20}$$

Note that here we are using a recursive equation, whereas in the main paper we used $X$ for input and $Y$ for output, to make things simple. Common choices of $\sigma(\cdot)$ are the Rectified Linear Unit (ReLU) activation function, the Leaky ReLU or the hyperbolic tangent function.

## A.2 MULTIPLE FEATURE GNNS

As mentioned earlier, the nodes of the graph are usually associated with a graph signal, which is multidimensional, i.e., $D > 1$ and $X^{(l)}$ is a matrix. Although the architecture in (20) can also handle multidimensional graph signals, multiple feature GNNs are commonly used, which are described by the following recursion formula:

$$X^{(l+1)} = \sigma \left( \sum_{k=0}^{K-1} S^k X^{(l)} H_k^{(l)} \right),\tag{21}$$

where $H_k^{(l)} \in \mathbb{R}^{F \times G}$ represents a set of $F \times G$ graph filters. Compared to the architecture in (20), the MIMO GNN employs multiple filters instead of one, and the outputs of the filters are combined to produce a layer output $X^{(l+1)}$ that has feature dimension equal to $G$.

## A.3 NOTATION

Our notation is summarized in Table 5.

## B RELATION TO OTHER ARCHITECTURES

GNNs have attracted significant attention and numerous architectures have been proposed. The first GNNs of (Scarselli et al., 2008b; Kipf & Welling, 2016; Battaglia et al., 2016; Defferrard et al., 2016)

Table 5: Overview of notation.

| | | |
|---|---|---|
| $\mathcal{G}$ | $\triangleq$ | Graph |
| $\mathcal{V}$ | $\triangleq$ | Set of nodes |
| $\mathcal{E}$ | $\triangleq$ | Set of edges |
| $\boldsymbol{S}$ | $\triangleq$ | $N \times N$ graph operator |
| $\boldsymbol{X}$ | $\triangleq$ | GNN input; represents the $N \times D$ matrix of node attributes (graph signal) |
| $\boldsymbol{x}$ | $\triangleq$ | GNN input; represents the vector of node attributes (graph signal) |
| $\boldsymbol{Z}$ | $\triangleq$ | matrix output of a linear filter |
| $\boldsymbol{z}$ | $\triangleq$ | vector output of a linear filter |
| $\boldsymbol{Y}$ | $\triangleq$ | matrix output of a GNN module; $\boldsymbol{Y} = \sigma\left(\boldsymbol{Z}\right)$ |
| $\boldsymbol{y}$ | $\triangleq$ | vector output of a GNN module; $\boldsymbol{y} = \sigma\left(\boldsymbol{z}\right)$ |
| $a$ | $\triangleq$ | scalar |
| $\boldsymbol{a}$ | $\triangleq$ | vector |
| $\boldsymbol{A}$ | $\triangleq$ | matrix |
| $\boldsymbol{A}^T$ | $\triangleq$ | transpose of matrix $\boldsymbol{A}$ |
| $\boldsymbol{A}_k$ | $\triangleq$ | $\boldsymbol{A}[k,:]^T$, $k$-th row of matrix $\boldsymbol{A}$ |
| $\boldsymbol{a}_k$ | $\triangleq$ | $\boldsymbol{A}[:,k]$, $k$-th column of matrix $\boldsymbol{A}$ |
| $\boldsymbol{U}$ | $\triangleq$ | eigenvector matrix |
| $\boldsymbol{U}[k,:]$ | $\triangleq$ | $k$-th row of $\boldsymbol{U}$ (row vector) |
| $\boldsymbol{U}[:,k]$ | $\triangleq$ | $k$-th column of $\boldsymbol{U}$ |
| $\boldsymbol{u}_k$ | $\triangleq$ | $k$-th eigenvector, $k$-th column of $\boldsymbol{U}$ |
| $\boldsymbol{I}$ | $\triangleq$ | Identity matrix |
| $\boldsymbol{1}$ | $\triangleq$ | vector of ones |
| $\boldsymbol{0}$ | $\triangleq$ | vector or matrix of zeros |
| $|\cdot|$ | $\triangleq$ | point-wise absolute value |
| $\binom{m}{n}$ | $\triangleq$ | binomial coefficient |

used simple convolutions in static data and graphs, whereas more sophisticated architectures utilize a variety of attention mechanisms (Hamilton et al., 2017; Veličković et al., 2018; Liu et al., 2021). Graph convolutional architectures have also been designed for time-varying graphs and signals. Some of them exploit both the graph and the time structure (Hajiramezanali et al., 2019; Wang et al., 2021; Hadou et al., 2021), while others employ recurrent architectures (Li et al., 2016; Seo et al., 2018; Nicolicioiu et al., 2019; Ruiz et al., 2020b).

It is often the case that GNNs are presented in the literature using different definitions. The GNN by (Kipf & Welling, 2016) for example is written as:

$$\boldsymbol{X}^{(l+1)} = \sigma\left(\boldsymbol{D}^{-1/2}\left(\boldsymbol{S}+\boldsymbol{I}\right)\boldsymbol{D}^{-1/2}\boldsymbol{X}^{(l)}\boldsymbol{H}^{(l)}\right) = \sigma\left(\boldsymbol{D}^{-1/2}\boldsymbol{S}\boldsymbol{D}^{-1/2}\boldsymbol{X}^{(l)}\boldsymbol{H}^{(l)} + \boldsymbol{D}^{-1}\boldsymbol{X}^{(l)}\boldsymbol{H}^{(l)}\right),$$
(22)

where $\boldsymbol{S} \in \{0,1\}^{N \times N}$ represents the graph adjacency, $\boldsymbol{D}$ is a diagonal matrix, with $\boldsymbol{D}[i,i]$ being the degree of node $i$. The matrix $\boldsymbol{D}^{-1/2}\left(\boldsymbol{S}+\boldsymbol{I}\right)\boldsymbol{D}^{-1/2}$ is also a GSO $\boldsymbol{S}'$ and the formula in (22) can be written as:

$$\boldsymbol{X}^{(l+1)} = \sigma\left(\boldsymbol{S}'\boldsymbol{X}^{(l)}\boldsymbol{H}^{(l)}\right),$$
(23)

which is a special case of the MIMO GNN in (21), for $K = 2$. Another way that GNNs are represented in the literature is via the following equations:

$$\boldsymbol{A}_v^{(l)} = \text{AGGREGATE}\left(\left\{\boldsymbol{X}_u^{(l)} : u \in \mathcal{N}(v)\right\}\right)$$
(24)

$$\boldsymbol{B}_v^{(l)} = \text{COMBINE}\left(\boldsymbol{X}_v^{(l)}, \boldsymbol{A}_v^{(l)}\right)$$
(25)

$$\boldsymbol{X}_v^{(l+1)} = \sigma\left(\boldsymbol{H}^{(l)}\boldsymbol{B}_v^{(l)}\right)$$
(26)

where $\boldsymbol{X}_v^{(l)}$ is the signal of node $v$ in layer $l$ and the $v$-th row of the feature matrix $\boldsymbol{X}^{(l)}$, i.e., $\boldsymbol{X}^{(l)} = \begin{bmatrix} \boldsymbol{X}_1^{(l)^T} \\ \vdots \\ \boldsymbol{X}_N^{(l)^T} \end{bmatrix}$. Equivalently, $\boldsymbol{A}_v^{(l)}, \boldsymbol{B}_v^{(l)}$ are rows of matrices $\boldsymbol{A}^{(l)}$, $\boldsymbol{B}^{(l)}$ respectively and represent signals associated with node $v$. The majority of the architectures based on the equations (24)-(26) can be written as combinations of the GNN modules in (21). Different architectures employ different functions for AGGREGATE and COMBINE. Popular choices of AGGREGATE functions include the mean, the sum, pooling functions or LSTM functions. The COMBINE routine, on the other hand, usually utilizes the concatanation or summation function. The settings that are mainly used are summation function for AGGREGATE and concatenation for COMBINE. This is due to the fact that summation-concatenation preserves the more information compared to other options Xu et al. (2019). It is then easy to see that:

$$\boldsymbol{A}^{(l)} = \boldsymbol{S}\boldsymbol{X}^{(l)} \tag{27}$$

$$\boldsymbol{B}^{(l)} = \left[\boldsymbol{A}^{(l)}, \boldsymbol{X}^{(l)}\right] \tag{28}$$

$$\boldsymbol{X}^{(l+1)} = \sigma\left(\boldsymbol{B}^{(l)}\boldsymbol{H}^{(l)}\right) = \sigma\left(\boldsymbol{S}\boldsymbol{X}^{(l)}\boldsymbol{H}_1^{(l)} + \boldsymbol{X}^{(l)}\boldsymbol{H}_0^{(l)}\right) = \sigma\left(\sum_{k=0}^{1}\boldsymbol{S}^k\boldsymbol{X}^{(l)}\boldsymbol{H}_k^{(l)}\right), \tag{29}$$

where $\boldsymbol{S}$ is the graph adjacency and $\boldsymbol{H}^{(l)} = \begin{bmatrix} \boldsymbol{H}_1^{(l)} \\ \boldsymbol{H}_0^{(l)} \end{bmatrix}$. Therefore, the GNN defined in (29) is a special case of the GNN in (21), for $K = 2$.

Now consider the GNN defined in (29) that consists of $K$ layers and $\sigma(\cdot)$ is the linear function for the hidden layers and a nonlinear activation function in the output layer, i.e.,

$$\boldsymbol{X}^{(l+1)} = \boldsymbol{S}\boldsymbol{X}^{(l)}\boldsymbol{H}^{(l)} + \boldsymbol{X}^{(l)}\boldsymbol{H}^{(l)} = (\boldsymbol{S} + \boldsymbol{I})\,\boldsymbol{X}^{(l)}\boldsymbol{H}^{(l)}, \text{ for } l = \{0, \ldots, K-2\} \tag{30}$$

$$\boldsymbol{X}^{(l+1)} = \sigma\left(\boldsymbol{S}\boldsymbol{X}^{(l)}\boldsymbol{H}^{(l)} + \boldsymbol{X}^{(l)}\boldsymbol{H}^{(l)}\right), \text{ for } l = K-1 \tag{31}$$

Then it holds that:

$$\boldsymbol{X}^{(l+1)} = (\boldsymbol{S} + \boldsymbol{I})^{l+1}\,\boldsymbol{X}^{(0)}\boldsymbol{H}^{(1)}\cdots\boldsymbol{H}^{(l)}, \text{ for } l = \{0, \ldots, K-2\} \tag{32}$$

$$\boldsymbol{X}^{(l+1)} = \sigma\left(\boldsymbol{S}\boldsymbol{X}^{(l)}\boldsymbol{H}^{(l)} + \boldsymbol{X}^{(l)}\boldsymbol{H}^{(l)}\right), \text{ for } l = K-1 \tag{33}$$

As a result:

$$\boldsymbol{X}^{(K)} = \sigma\left((\boldsymbol{S} + \boldsymbol{I})^K\,\boldsymbol{X}^{(0)}\boldsymbol{H}^{(K-1)}\cdots\boldsymbol{H}^{(0)}\right) = \sigma\left(\sum_{l=0}^{K}\boldsymbol{S}^l\boldsymbol{X}^{(0)}\boldsymbol{H}_l'\right), \tag{34}$$

which again corresponds to the GNN in (21). The last equality holds since

$$(\boldsymbol{X} + \boldsymbol{I})^K = \sum_{l=0}^{K}\binom{K}{l}\boldsymbol{S}^{K-l}, \tag{35}$$

where $\binom{n}{k} = \frac{n!}{k!(n-k)!}$ is the binomial coefficient. Overall there is a direct connection between the GNNs defined by the equations (24)-(26) and the GNNs defined by (21). Furthermore, apropriate selection of GSO and nonlinearities in (21) with respect to the AGGREGATE and COMBINE functions in (24)-(26) makes the described architectures equivalent.

## C ASSOCIATING THE WL ALGORITHM WITH THE SPECTRAL DECOMPOSITION OF A GRAPH

In section 3 we observed a connection between the limitations of the WL algorithm and graphs with eigenvectors orthogonal to $\mathbf{1}$. The WL algorithm is initialized with either $\boldsymbol{x} = \mathbf{1}$ or $\boldsymbol{x} = \boldsymbol{S}\mathbf{1}$ and in

the remaining iterations this information is propagated (diffused) through the nodes. In particular, at iteration $k$ of the WL algorithm, node $i$ receives a multiset defined as:

$$\mathcal{T}_i^k : \left\{ x_j \in \mathcal{T}_i^k | x_j = \sum_n \lambda_n \left( \boldsymbol{u}_n^T \mathbf{1} \right) \boldsymbol{u}_n(j), \quad j \in \mathcal{N}_i^k \right\}, \tag{36}$$

where $\mathcal{N}_i^k$ denotes the $k-$th neighborhood of node $i$. If there is one-to-one correspondence between $\mathcal{T}_i^k$ and $\boldsymbol{S}^k \mathbf{1}(i)$ for all nodes $i$ then the WL algorithm can be analyzed by building and comparing the following features for each node:

$$\boldsymbol{X} = \left[ \boldsymbol{S}\mathbf{1}, \boldsymbol{S}^2\mathbf{1}, \dots, \boldsymbol{S}^K\mathbf{1} \right] \tag{37}$$

In other words, if the summation operation is a proper hash function for a specific graph, the WL algorithm is equivalent to the feature generation of (37). In that case, we can use the spectral decomposition of $\boldsymbol{S}$ and the analysis of section 3 to characterize the limitations of the WL algorithm. Then the WL algorithm admits the same limitations as the GNNs with $\boldsymbol{x} = \mathbf{1}$ input and it omits the information associated with eigenvectors that are orthogonal to $\mathbf{1}$.

## D  THE STOCHASTIC GNN MODULE

In this section we elaborate more on the proposed stochastic GNN module in Fig. 3a. In order to implement it, we can either use the equivalent model in Fig. 3b or we can design an empirical variance model. In practice, the input to the empirical model is a matrix $\boldsymbol{X} \in \mathbb{R}^{N \times M}$ where each element is independently drawn from a Gaussian distribution with zero mean and unit variance and $M$ is the total number of samples. The output of the filter is $\boldsymbol{Z} = \sum_k h_k \boldsymbol{S}^k \boldsymbol{X} \in \mathbb{R}^{N \times M}$ and the maximum likelihood estimate of the empirical covariance of $\boldsymbol{Z}$ takes the form:

$$\boldsymbol{Q} = \frac{1}{M} \boldsymbol{Z}\boldsymbol{Z}^T. \tag{38}$$

Then the GNN output can be written as:

$$\boldsymbol{y} = \mathrm{diag}\left(\boldsymbol{Q}\right) = \frac{1}{M} \mathrm{diag}\left(\boldsymbol{Z}\boldsymbol{Z}^T\right) = \frac{1}{M}\boldsymbol{Z}^2\mathbf{1} \tag{39}$$

The GNN module of the empirical variance model is illustrated in Fig. 6.

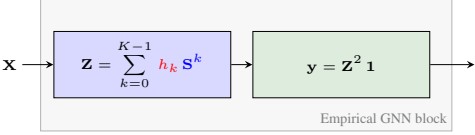

Figure 6: Empirical variance GNN module

### D.1  THE EFFECT OF SQUARE NONLINEARITY

Now we discuss the effect of square nonlinearity in the representation of nodes, introduced by the variance operator. As mentioned in section 4 the nonlinearity added by the variance computation allows the proposed GNN to gather information from farther neighborhoods compared to a linear filter or the WL algorithm. To make things more concrete, consider the graph in Fig. 6 and let $K - 1 = 2$, which corresponds to running the WL algorithm for 2 iterations and graph filters that process $\boldsymbol{S}$ and $\boldsymbol{S}^2$.

In Table 7 we present the representations produced by the stochastic GNN and the WL algorithm for each node of the graph in Fig. 6. In particular, we present two iterations of the WL algorithm and the value that $\boldsymbol{y}$ in (39) converges to, for filter values $(h_0, h_1, h_2) = (3, 5, 7)$. We observe that the WL algorithm represents nodes A and D with the same value, whereas the output of the stochastic GNN is capable of separating these two nodes. Overall, the the nonlinearity in the variance operator allows acquiring global information, which can be vital in the resulting node representation of the graph.

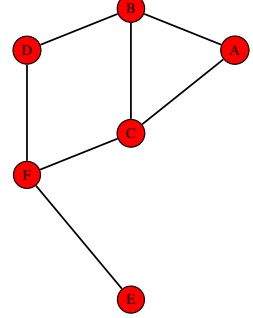

Table 6: graph

Table 7: GNN vs WL algorithm on the graph in Fig. 6 for $K = 3$.

| NODE | WL ALGORITHM | GNN |
|------|--------------|-----|
| A | 2, 3 3 | 675 |
| B | 3, 2 2 3 | 1085 |
| C | 3, 2 3 3 | 1134 |
| D | 2, 3 3 | 633 |
| E | 1, 3 | 223 |
| F | 3, 1 2 3 | 896 |

### D.2 COMPUTING THE COVARIANCE RECURSIVELY

In the main core of the paper we presented 3 almost eguivalent GNN modules; the stochastic module with random Gaussian input, the diagonal module with no input and the standard GNN module with input designed by (14). On the basis of the requirements and constraints of each task, we can employ either of them in a GNN architecture. For instance, in applications where computing the adjacency power diagonals is computationally prohibitive, we can use the empirical module in Fig. 6. The drawback is that for systems with high variance, a significant number of samples will be required for the output to converge. This can be mitigated by computing the output in (39) recursively. To be more precise, let $\boldsymbol{z}_m$ be the $m-$th column of the filter output $\boldsymbol{Z}$ and $\boldsymbol{Q}_{(M)}$, $\boldsymbol{y}_{(M)}$ be the empirical covariance and output after obtaining $M$ samples. Then the recursive equations can be written as:

$$\boldsymbol{Q}_{(M)} = \frac{1}{M}\sum_{m=1}^{M} \boldsymbol{z}_m \boldsymbol{z}_m^T = \frac{1}{M}\sum_{m=1}^{M-1} \boldsymbol{z}_m \boldsymbol{z}_m^T + \frac{1}{M}\boldsymbol{z}_M \boldsymbol{z}_M^T = \frac{M-1}{M}\boldsymbol{Q}_{(M-1)} + \frac{1}{M}\boldsymbol{z}_M \boldsymbol{z}_M^T \quad (40)$$

$$\boldsymbol{y}_{(M)} = \text{diag}\left(\boldsymbol{Q}_{(M)}\right) = \frac{M-1}{M}\text{diag}\left(\boldsymbol{Q}_{(M-1)}\right) + \frac{1}{M}\text{diag}\left(\boldsymbol{z}_M \boldsymbol{z}_M^T\right) = \frac{M-1}{M}\boldsymbol{y}_{(M-1)} + \frac{1}{M}|\boldsymbol{z}_M|^2 \quad (41)$$

Therefore, using $\boldsymbol{y}_{(M)} = \frac{M-1}{M}\boldsymbol{y}_{(M-1)} + \frac{1}{M}|\boldsymbol{z}_M|^2$, allows for online computations and reduces the required memory complexity.

## E PROOF OF THEOREM 2.2

To prove Theorem 2.2, consider the GNN module in (1), where $\boldsymbol{H}_k$ is a scalar, that is,

$$\boldsymbol{Y} = \sigma\left(\sum_{k=0}^{K-1} h_k \boldsymbol{S}^k \boldsymbol{X}\right) \quad (42)$$

### E.1 CASE 1: THERE DOES NOT EXIST PERMUTATION MATRIX $\boldsymbol{\Pi}$ SUCH THAT $\boldsymbol{X} = \boldsymbol{\Pi}\hat{\boldsymbol{X}}$.

Consider an $1-$ layer GNN with 2 neurons defined by $h_0 = 1$, $h_i = 0$, $i \neq 0$ and $h_0 = -1$, $h_i = 0$, $i \neq 0$, i.e.,

$$\boldsymbol{Y}_1 = \sigma\left(\boldsymbol{X}\right), \quad \boldsymbol{Y}_2 = \sigma\left(-\boldsymbol{X}\right) \quad (43)$$

Summing up the output of the 2 neurons to produce the final GNN output yields $\boldsymbol{Y} = \boldsymbol{Y}_1 + \boldsymbol{Y}_2 = \boldsymbol{X}$ when the $\sigma(\cdot) = \text{ReLU}(\cdot)$. As a result, the output of the GNN is the graph signal and since there does not exist permutation matrix $\boldsymbol{\Pi}$ such that $\boldsymbol{X} = \boldsymbol{\Pi}\hat{\boldsymbol{X}}$, this GNN decides that $\mathcal{G}$ and $\hat{\mathcal{G}}$ are different.

### E.2 CASE 2: THERE EXISTS $\lambda \in \mathcal{S}$, SUCH THAT $\lambda \notin \hat{\mathcal{S}}$ AND $\boldsymbol{X}^T \boldsymbol{V}_\lambda \neq \boldsymbol{0}$.

Let $\mathcal{S} = \{\lambda_1, \ldots, \lambda_p\}$ be the set containing the unique (non-repeated) eigenvalues of $\boldsymbol{S}$ and $\hat{\mathcal{S}} = \{\hat{\lambda}_1, \ldots, \hat{\lambda}_r\}$ be the set containing the unique eigenvalues of $\hat{\boldsymbol{S}}$. Note that the eigenvalues of $\boldsymbol{S}$, $\hat{\boldsymbol{S}}$

are not required to be distinct. Also, let $\{\mu_1, \ldots, \mu_q\}$ be the set of all distinct eigenvalues of $\boldsymbol{S}$ and $\hat{\boldsymbol{S}}$, i.e., $\mu_i \in \mathcal{S} \bigcup \hat{\mathcal{S}}$ and $\mu_i \neq \mu_j$, $\forall\, i \neq j$. Suppose that $\boldsymbol{S}, \ \hat{\boldsymbol{S}}$ have at least one different eigenvalue, i.e., there exists $\lambda \in \mathcal{S}$ such that $\lambda \notin \hat{\mathcal{S}}$.

Recall from Appendix A that a graph filter can be represented in the frequency domain by:

$$\tilde{\boldsymbol{H}}\left(\lambda_i\right) = \sum_{k=0}^{K-1} h_k \lambda_i^k, \tag{44}$$

Then:

$$\begin{bmatrix} \tilde{\boldsymbol{H}}\left(\mu_1\right) \\ \tilde{\boldsymbol{H}}\left(\mu_2\right) \\ \vdots \\ \tilde{\boldsymbol{H}}\left(\mu_q\right) \end{bmatrix} = \begin{bmatrix} 1\ \mu_1\ \mu_1^2 \ldots \mu_1^{K-1} \\ 1\ \mu_2\ \mu_2^2 \ldots \mu_2^{K-1} \\ \vdots \\ 1\ \mu_q\ \mu_q^2 \ldots \mu_q^{K-1} \end{bmatrix} \begin{bmatrix} h_0 \\ h_1 \\ \vdots \\ h_{K-1} \end{bmatrix} = \boldsymbol{W}\boldsymbol{h} \tag{45}$$

$\boldsymbol{W}$ is a Vandermonde matrix and when $K = q$ the determinant of $\boldsymbol{W}$ takes the form:

$$\det\left(\boldsymbol{W}\right) = \boldsymbol{\Pi}_{1 \leq i < j \leq q}\left(\mu_i - \mu_j\right) \tag{46}$$

Since the values $\mu_i$ are distinct, $\boldsymbol{W}$ has full column rank and there exists a graph filter $\boldsymbol{H}\left(\cdot\right)$ with unique parameters $\boldsymbol{h}$ that passes only the $\lambda$ eigenvalue, i.e.,

$$\tilde{\boldsymbol{H}}\left(\mu_i\right) = \begin{cases} 1, & \text{if } \mu_i = \lambda \\ 0, & \text{if } \mu_i \neq \lambda \end{cases} \tag{47}$$

Under this parametrization, the filter $\boldsymbol{H}\left(\cdot\right)$ takes the form $\boldsymbol{H}\left(\boldsymbol{S}\right) = \boldsymbol{V}_\lambda \boldsymbol{V}_\lambda^T$, where $\boldsymbol{V}_\lambda$ is the eigenspace (orthogonal space of the eigenvectors) corresponding to $\lambda$, and $\boldsymbol{H}\left(\hat{\boldsymbol{S}}\right) = 0$. Then the output of the GNN, for the two graphs, takes the form:

$$\boldsymbol{Y} = \sigma\left(\boldsymbol{H}\left(\boldsymbol{S}\right)\boldsymbol{X}\right) = \sigma\left(\boldsymbol{V}_\lambda \boldsymbol{V}_\lambda^T \boldsymbol{X}\right) \tag{48}$$

$$\hat{\boldsymbol{Y}} = \sigma\left(\boldsymbol{H}\left(\hat{\boldsymbol{S}}\right)\hat{\boldsymbol{X}}\right) = \boldsymbol{0} \tag{49}$$

Under the assumption that $\boldsymbol{X}^T \boldsymbol{V}_\lambda \neq \boldsymbol{0}$, we also have $\boldsymbol{V}_\lambda \boldsymbol{V}_\lambda^T \boldsymbol{X} \neq \boldsymbol{0}$. As a result $\sigma\left(\boldsymbol{V}_\lambda \boldsymbol{V}_\lambda^T \boldsymbol{X}\right) \neq \boldsymbol{0}$, there does not exist a permutation $\boldsymbol{\Pi}$ such that $\boldsymbol{Y} = \boldsymbol{\Pi}\hat{\boldsymbol{Y}}$ and the proposed GNN decides that the two graphs are different. Note $\sigma\left(\boldsymbol{V}_\lambda \boldsymbol{V}_\lambda^T \boldsymbol{X}\right) \neq \boldsymbol{0}$ always holds when, for example, leaky ReLU is used that allows both positive and negative values to pass. In the case where $\sigma\left(\cdot\right) = \text{ReLU}\left(\cdot\right)$ the proof is still valid as long as there is at least one positive value in $\boldsymbol{V}_\lambda \boldsymbol{V}_\lambda^T \boldsymbol{X}$. In case $\boldsymbol{V}_\lambda \boldsymbol{V}_\lambda^T \boldsymbol{X} \leq \boldsymbol{0}$ we can without loss of generality consider the filter:

$$\tilde{\boldsymbol{H}}\left(\mu_i\right) = \begin{cases} -1, & \text{if } \mu_i = \lambda \\ 0, & \text{if } \mu_i \neq \lambda \end{cases} \tag{50}$$

that results in $\sigma\left(-\boldsymbol{V}_\lambda \boldsymbol{V}_\lambda^T \boldsymbol{X}\right) \neq \boldsymbol{0}$.

## E.3 CASE 3: THERE EXISTS $\lambda \in \mathcal{S}, \hat{\mathcal{S}}$, SUCH THAT $m_\lambda \neq \hat{m}_\lambda$ AND $\boldsymbol{X}^T\left(\boldsymbol{V}_\lambda \oplus \hat{\boldsymbol{V}}_\lambda\right) \neq \boldsymbol{0}$.

Let $\mathcal{S} = \{\lambda_1, \ldots, \lambda_p\}$ be the set containing the unique (non-repeated) eigenvalues of $\boldsymbol{S}$ with multiplicities $\{m_{\lambda_1}, \ldots, m_{\lambda_p}\}$ and $\hat{\mathcal{S}} = \{\hat{\lambda}_1, \ldots, \hat{\lambda}_r\}$ be the set containing the unique eigenvalues of $\hat{\boldsymbol{S}}$ with multiplicities $\{\hat{m}_{\hat{\lambda}_1}, \ldots, \hat{m}_{\hat{\lambda}_r}\}$. Note that the eigenvalues of $\boldsymbol{S}, \ \hat{\boldsymbol{S}}$ are not required to be distinct. Also, let $\{\mu_1, \ldots, \mu_q\}$ be the set of all distinct eigenvalues of $\boldsymbol{S}$ and $\hat{\boldsymbol{S}}$, i.e., $\mu_i \in \mathcal{S} \bigcup \hat{\mathcal{S}}$ and $\mu_i \neq \mu_j$, $\forall\, i \neq j$. Suppose that $\boldsymbol{S}, \ \hat{\boldsymbol{S}}$ have at least one common eigenvalue but with different multiplicity, i.e., there exists $\lambda \in \mathcal{S}, \hat{\mathcal{S}}$, such that $m_\lambda \neq \hat{m}_\lambda$.

Under the parametrization of (47), $\boldsymbol{H}\left(\boldsymbol{S}\right) = \boldsymbol{V}_\lambda \boldsymbol{V}_\lambda^T$, where $\boldsymbol{V}_\lambda \in \mathbb{R}^{N \times m_\lambda}$ is the eigenspace of $\boldsymbol{S}$ corresponding to $\lambda$, and $\boldsymbol{H}\left(\hat{\boldsymbol{S}}\right) = \hat{\boldsymbol{V}}_\lambda \hat{\boldsymbol{V}}_\lambda^T$, where $\hat{\boldsymbol{V}}_\lambda \in \mathbb{R}^{N \times \hat{m}_\lambda}$ is the eigenspace of $\hat{\boldsymbol{S}}$

corresponding to $\lambda$. Then the output of the GNN, for the two graphs, takes the form:

$$Y = \sigma\left(H\left(S\right)X\right) = \sigma\left(V_\lambda V_\lambda^T X\right) \tag{51}$$

$$\hat{Y} = \sigma\left(H\left(\hat{S}\right)\hat{X}\right) = \sigma\left(\hat{V}_\lambda \hat{V}_\lambda^T X\right) \tag{52}$$

The subspaces $V_\lambda$, $\hat{V}_\lambda$ can be written as:

$$V_\lambda = [Q_c, Q_n]\,W,\ \hat{V}_\lambda = \left[Q_c, \hat{Q}_n\right]\hat{W}, \tag{53}$$

where $Q_c \in \mathbb{R}^{N\times c}$ is the common subspace between $V_\lambda$ and $\hat{V}_\lambda$ and $Q_n \in \mathbb{R}^{N\times(m_\lambda-c)}$, $\hat{Q}_n \in \mathbb{R}^{N\times(\hat{m}_\lambda-c)}$ are subspaces of $V_\lambda$ and $\hat{V}_\lambda$ such that $Q_n \bigcap \hat{Q}_n = \{0\}$. Since $m_\lambda \neq \hat{m}_\lambda$, $Q_n + \hat{Q}_n \neq \{0\}$, where $+$ denotes here the sum between subspaces. Furthemore $W \in \mathbb{R}^{m_\lambda\times m_\lambda}$, $\hat{W} \in \mathbb{R}^{\hat{m}_\lambda\times\hat{m}_\lambda}$ are square orthogonal matrices.

As a result

$$Y = \sigma\left(V_\lambda V_\lambda^T X\right) = \sigma\left([Q_c, Q_n]\,WW^T\,[Q_c, Q_n]^T X\right) = \sigma\left(Q_c Q_c^T X + Q_n Q_n^T X\right) \tag{54}$$

$$\hat{Y} = \sigma\left(\hat{V}_\lambda \hat{V}_\lambda^T X\right) = \sigma\left(\left[Q_c, \hat{Q}_n\right]\hat{W}\hat{W}^T\left[Q_c, \hat{Q}_n\right]^T X\right) = \sigma\left(Q_c Q_c^T X + \hat{Q}_n \hat{Q}_n^T X\right) \tag{55}$$

We define:

$$V_\lambda \oplus \hat{V}_\lambda := \{u + w \mid u \in V_\lambda; w \in \hat{V}_\lambda; u, w \notin V_\lambda \bigcap \hat{V}_\lambda\}, \tag{56}$$

as the exclusive sum of subspace. If $X^T\left(V_\lambda \oplus \hat{V}_\lambda\right) \neq 0$ then either $Q_n^T X \neq 0$ or $\hat{Q}_n^T X \neq 0$. Therefore, there is no permutation $\Pi$ such that $V_\lambda V_\lambda^T X = \Pi \hat{V}_\lambda \hat{V}_\lambda^T X$ since there is no permutation that makes these vectors collinear. Our previous analysis shows that suitable nonlinearities will also guarantee that there is no permutation $\Pi$ such that $Y = \Pi\hat{Y}$.

## F   PROOF OF THEOREMS 4.1, 5.2, 6.1:

The proof of Theorems 5.2, 6.1 is equivalent and very similar to the proof of Theorem 4.1. We begin by proving Theorem 5.2. To prove Theorem 5.2 let us consider again the GNN module in (10).

$$y = \sigma\left(\sum_{k=0}^{K-1} h_k \mathrm{diag}\left(S^k\right)\right). \tag{57}$$

For simplicity we assume that $\sigma\left(\cdot\right)$ is a linear function. In eq. (43) we show how to produce the linear function from ReLU. If Assumption 2.1 holds, there exists $\lambda \in \mathcal{S}$, such that $\lambda \notin \hat{\mathcal{S}}$ or $m_\lambda \neq \hat{m}_\lambda$. We use the proof of Theorem 2.2 and conclude that there exists a graph filter $H\left(\cdot\right)$ with unique parameters $h$ that passes only the $\lambda$ eigenvalue, i.e.,

$$\tilde{H}\left(\mu_i\right) = \begin{cases} 1, & \text{if } \mu_i = \lambda \\ 0, & \text{if } \mu_i \neq \lambda \end{cases} \tag{58}$$

First we study the case where $\lambda \in \mathcal{S}$, but $\lambda \notin \hat{\mathcal{S}}$. Then $H\left(S\right) = V_\lambda V_\lambda^T$, where $V_\lambda \in \mathbb{R}^{N\times m_\lambda}$ is the eigenspace of $S$ corresponding to $\lambda$, and $H\left(\hat{S}\right) = 0$. The output $y$ of (57), for the two graphs, takes the form:

$$y = \mathrm{diag}\left(H\left(S\right)\right) = |V_\lambda[:,1]|^2 + \cdots + |V_\lambda[:,m]|^2 = \sum_{i=1}^m |V_\lambda[:,i]|^2 \tag{59}$$

$$\hat{y} = \mathrm{diag}\left(H\left(\hat{S}\right)\right) = 0 \tag{60}$$

where $V_\lambda[:, i]$ is the $i-$th column of $V_\lambda$. Since $V_\lambda \neq 0$ by definition, there does not exist a permutation $\Pi$ such that $y = \Pi \hat{y}$ and the proposed GNN can tell the two graphs apart.

Next we study the case where $\lambda \in \mathcal{S}, \hat{\mathcal{S}}$, but $m_\lambda \neq \hat{m}_\lambda$. Then $H(S) = V_\lambda V_\lambda^T$ and $H\left(\hat{S}\right) = \hat{V}_\lambda \hat{V}_\lambda^T$, where $\hat{V}_\lambda \in \mathbb{R}^{N \times \hat{m}_\lambda}$ is the eigenspace of $\hat{S}$ corresponding to $\lambda$. The output $y$ of (57), for the two graphs, takes the form:

$$y = \text{diag}\left(H\left(S\right)\right) = \sum_{i=1}^{m} |V_\lambda[:, i]|^2 \tag{61}$$

$$\hat{y} = \text{diag}\left(H\left(\hat{S}\right)\right) = \sum_{i=1}^{m} |\hat{V}_\lambda[:, i]|^2 \tag{62}$$

We observe that:

$$\mathbf{1}^T y = \mathbf{1}^T \text{diag}\left(H\left(S\right)\right) = \text{Trace}\left(V_\lambda V_\lambda^T\right) = m_\lambda \tag{63}$$

$$\mathbf{1}^T \hat{y} = \mathbf{1}^T \text{diag}\left(H\left(\hat{S}\right)\right) = \text{Trace}\left(\hat{V}_\lambda \hat{V}_\lambda^T\right) = \hat{m}_\lambda \tag{64}$$

Since $m_\lambda \neq \hat{m}_\lambda$, there is no permutation $\Pi$ such that $y = \Pi \hat{y}$ and the proposed GNN can tell the two graphs apart. This concludes the proof for Theorem 5.2. Using Proposition 6.1 we prove the equivalence of Theorems 5.2 and 6.1 and therefore the proof is the same.

To prove Theorem 4.1 we need one more extra step. In particular, we plug the filter, with parametrization as in (58), in equation (8), for $S, \hat{S}$, i.e.,

$$\text{cov}\left[z; S\right] = \sum_{k=0}^{K-1} h_k S^k \sum_{m=0}^{K-1} h_m S^m = V_\lambda V_\lambda^T V_\lambda V_\lambda^T = V_\lambda V_\lambda^T \tag{65}$$

$$\text{cov}\left[z; \hat{S}\right] = \sum_{k=0}^{K-1} h_k \hat{S}^k \sum_{m=0}^{K-1} h_m \hat{S}^m = \hat{V}_\lambda \hat{V}_\lambda^T \hat{V}_\lambda \hat{V}_\lambda^T = \hat{V}_\lambda \hat{V}_\lambda^T, \tag{66}$$

where the last equality in (65) holds, since $V_\lambda, \hat{V}_\lambda$ are orthogonal. Then the output $y$ of (9), for the two graphs, can be written as:

$$y = \text{var}\left[z; S\right] = \text{diag}\left(\text{cov}\left[z; S\right]\right) = |V_\lambda[:, 1]|^2 + \cdots + |V_\lambda[:, m]|^2 = \sum_{i=1}^{m} |V_\lambda[:, i]|^2 \tag{67}$$

$$\hat{y} = \text{var}\left[z; S\right] = \text{diag}\left(\text{cov}\left[z; S\right]\right) = |\hat{V}_\lambda[:, 1]|^2 + \cdots + |\hat{V}_\lambda[:, m]|^2 = \sum_{i=1}^{m} |\hat{V}_\lambda[:, i]|^2 \tag{68}$$

If $\lambda \in \mathcal{S}$ but $\lambda \notin \hat{\mathcal{S}}$, $\hat{y} = 0$. If $\lambda \in \mathcal{S}, \hat{\mathcal{S}}$, but $m_\lambda \neq \hat{m}_\lambda$, $\mathbf{1}^T y \neq \mathbf{1}^T \hat{y}$. In any case, there does not exist a permutation $\Pi$ such that $y = \Pi \hat{y}$ and the proposed stochastic GNN can separate the two graphs.

## G   PROOF OF PROPOSITION 6.1

The output of type-1 module can be cast as:

$$y = \sigma\left(\sum_{k=0}^{K-1} h_k \text{diag}\left(S^k\right)\right) = \sigma\left(X h\right), \tag{69}$$

when $X$ is designed as in (14) and $h = \begin{bmatrix} h_0 \\ \vdots \\ h_{K-1} \end{bmatrix}$ is the vector of filter parameters. The same output can be produced by the type-2 module when $H_k$ is a vector and $K = 1$. On the other hand, a set of

$K$ type-1 modules in the input layer can produce the $\boldsymbol{X}$ in (14). To see this, consider the following type-1 GNN modules.

$$\boldsymbol{y}_i = \sigma \left( \sum_{k=0}^{K-1} h_k^{(i)} \mathrm{diag}\left(\boldsymbol{S}^k\right) \right), \ \ i = 0, \ldots, K-1, \tag{70}$$

where

$$h_k^{(i)} = \begin{cases} 1, \ \text{if} \ i = k \\ 0, \ \text{if} \ i \neq k \end{cases} \tag{71}$$

Concatenating the outputs $\boldsymbol{y}_i$ into $\boldsymbol{W} = [\boldsymbol{y}_0, \ldots, \boldsymbol{y}_{K-1}]$ results in the $\boldsymbol{X}$ in (14) which we can apply to a type-2 module and produce the same output as a type-2 GNN module with input as in (14). $\square$

## H   NONISOMORPHIC GRAPHS WITH THE SAME SET OF EIGENVALUES

In the core of this paper, we discuss the ability of GNNs to distinguish between nonisomorphic graphs that have different eigenvalues. This analysis covers the majority of real graphs, since real graphs almost never share the same eigenvalues. However, there exist interesting cases of graphs with the same set of eigenvalues that GNNs can also distinguish. In this section, we study these cases and provide interesting results.

### H.1   GRAPHS WITH THE SAME DISTINCT EIGENVALUES

We consider the case where $\boldsymbol{S}$, $\hat{\boldsymbol{S}}$ have distinct eigenvalues which are the same, i.e., $\boldsymbol{\Lambda} = \hat{\boldsymbol{\Lambda}}$. Formally:

**Assumption H.1** $\boldsymbol{S}$, $\hat{\boldsymbol{S}}$ *have the same distinct eigenvalues, i.e., $\mathcal{S} \subseteq \hat{\mathcal{S}}$ and $\hat{\mathcal{S}} \subseteq \mathcal{S}$, with $\lambda_i \neq \lambda_j$ for all $i$, $j$.*

Lemma H.2 characterizes nonisomorphic graphs with distinct eigenvalues.

**Lemma H.2** *When $\boldsymbol{S}$, $\hat{\boldsymbol{S}}$ have the same distinct eigenvalues, $\mathcal{G}$, $\hat{\mathcal{G}}$ are nonisomorphic if and only if there is no permutation matrix $\boldsymbol{\Pi}$ and diagonal $\pm 1$ matrix $\boldsymbol{D}$ such that:*

$$\boldsymbol{U} = \boldsymbol{\Pi}\hat{\boldsymbol{U}}\boldsymbol{D}$$

*Proof:* Let $\boldsymbol{S} = \boldsymbol{U}\boldsymbol{\Lambda}\boldsymbol{U}^T$, $\hat{\boldsymbol{S}} = \hat{\boldsymbol{U}}\hat{\boldsymbol{\Lambda}}\hat{\boldsymbol{U}}^T$. Since $\boldsymbol{S}$, $\hat{\boldsymbol{S}}$ have the same distinct eigenvalues, we have $\boldsymbol{\Lambda} = \hat{\boldsymbol{\Lambda}}$. To prove the 'forward' statement assume that $\mathcal{G}$, $\hat{\mathcal{G}}$ are nonisomorphic, i.e., there does not exist permutation matrix $\boldsymbol{\Pi}$ such that $\boldsymbol{S} = \boldsymbol{\Pi}\hat{\boldsymbol{S}}\boldsymbol{\Pi}^T$. If there exist permutation matrix $\boldsymbol{\Pi}$ and $\pm 1$ diagonal matrix $D$ such that $\boldsymbol{U} = \boldsymbol{\Pi}\hat{\boldsymbol{U}}\boldsymbol{D}$, then:

$$\boldsymbol{S} = \boldsymbol{U}\boldsymbol{\Lambda}\boldsymbol{U}^T = \boldsymbol{\Pi}\hat{\boldsymbol{U}}\boldsymbol{D}\boldsymbol{\Lambda}\boldsymbol{D}\hat{\boldsymbol{U}}^T\boldsymbol{\Pi}^T = \boldsymbol{\Pi}\hat{\boldsymbol{U}}\hat{\boldsymbol{\Lambda}}\hat{\boldsymbol{U}}^T\boldsymbol{\Pi}^T = \boldsymbol{\Pi}\hat{\boldsymbol{S}}\boldsymbol{\Pi}^T.$$

By contradiction when $\boldsymbol{S}$, $\hat{\boldsymbol{S}}$ have the same distinct eigenvalues, $\mathcal{G}$, $\hat{\mathcal{G}}$ are nonisomorphic if there do not exist a permutation matrix $\boldsymbol{\Pi}$ and a diagonal $\pm 1$ matrix $\boldsymbol{D}$ such that $\boldsymbol{U} = \boldsymbol{\Pi}\hat{\boldsymbol{U}}\boldsymbol{D}$.

To prove the 'backward' statement assume that there do not exist permutation matrix $\boldsymbol{\Pi}$ and diagonal $\pm 1$ matrix $\boldsymbol{D}$ such that $\boldsymbol{U} = \boldsymbol{\Pi}\hat{\boldsymbol{U}}\boldsymbol{D}$. If $\mathcal{G}$, $\hat{\mathcal{G}}$ are isomorphic, i.e., there exists permutation matrix $\boldsymbol{\Pi}$ such that $\boldsymbol{S} = \boldsymbol{\Pi}\hat{\boldsymbol{S}}\boldsymbol{\Pi}^T$, then:

$$\boldsymbol{U}\boldsymbol{\Lambda}\boldsymbol{U}^T = \boldsymbol{\Pi}\hat{\boldsymbol{U}}\boldsymbol{\Lambda}\hat{\boldsymbol{U}}^T\boldsymbol{\Pi}^T,$$

which implies that $\boldsymbol{u}_n = \pm\boldsymbol{\Pi}\hat{\boldsymbol{u}}_n$ for all $n$, where $\boldsymbol{u}_n$, $\hat{\boldsymbol{u}}_n$ refer to the columns of $\boldsymbol{U}$, $\hat{\boldsymbol{U}}$ respectively. As a result, $\boldsymbol{U} = \boldsymbol{\Pi}\hat{\boldsymbol{U}}\boldsymbol{D}$ and by contradiction we prove the 'backward' statement which concludes the proof. $\square$

In a nutshell, Lemma H.2 states that in order for $\mathcal{G}$, $\hat{\mathcal{G}}$ to be nonisomorphic, while Assumption H.1 holds, the two graphs need to admit different eigenvectors that correspond to the same eigenvalues. As a side note, we mention that $\boldsymbol{S}$, $\hat{\boldsymbol{S}}$ can still span the same columnspace, under row permutation. However, the power on each eigendirection has to be different for them to be nonisomorphic.

We can now extend the results of Theorem 2.2 to the following:

**Theorem H.3** *Let $\mathcal{G}$, $\hat{\mathcal{G}}$ be nonisomorphic graphs with graph signals $\boldsymbol{X}$, $\hat{\boldsymbol{X}}$. There exists a GNN that tells $\mathcal{G}$ and $\hat{\mathcal{G}}$ apart if:*

1. *There does not exist permutation matrix $\boldsymbol{\Pi}$ such that $\boldsymbol{X} = \boldsymbol{\Pi}\hat{\boldsymbol{X}}$, or*

2. *Assumption 2.1 holds and $\boldsymbol{V}_\lambda^T \boldsymbol{X} \neq \boldsymbol{0}$, or*

3. *Assumption H.1 holds and $\boldsymbol{X}^T \boldsymbol{u}_n \neq \boldsymbol{0}$ for all eigenvectors $\boldsymbol{u}_n$ or $\hat{\boldsymbol{X}}^T \hat{\boldsymbol{u}}_n \neq \boldsymbol{0}$ for all eigenvectors $\hat{\boldsymbol{u}}_n$.*

*Proof:* The proof for cases 1 and 2 can be found in Appendix E. Case 3 includes Assumption H.1, i.e., both $\boldsymbol{S}$, $\hat{\boldsymbol{S}}$ have $N$ distinct eigenvalues, where $N$ is the number of nodes in each graph, and also $\boldsymbol{S}$, $\hat{\boldsymbol{S}}$ share the same eigenvalues. To prove this last part of Theorem 2.2 we consider an $1-$layer GNN with $N$ neurons. Each neuron consists of a graph filter that isolates one eigenvalue and sets it to one, as in Appendix E. Then, each neuron is described by the following set of equations:

$$\boldsymbol{Y}_n = \sigma\left(\boldsymbol{H}_n\left(\boldsymbol{S}\right)\boldsymbol{X}\right), \quad n = 1, \ldots, N \tag{72}$$

$$\tilde{\boldsymbol{H}}_n\left(\lambda_i\right) = \begin{cases} 1, & \text{if } i = n \\ 0, & \text{if } i \neq n \end{cases}, \quad n = 1, \ldots, N \tag{73}$$

For the rest of the proof, we will assume that $\sigma(\cdot)$ is the linear function. This is without loss of generality since if we double the number of neurons in the layer and set $\sigma(\cdot) = \text{ReLU}(\cdot)$ we can produce the same output as the linear function by using the same trick as in Appendix E.1. In particular, $N$ of the graph filters will follow the equations in (73) and the remaining $N$ filters will follow the same equation with $-1$ instead, as in (50). Then for each eigenvalue we have a pair of filters, one with $+1$ and one with $-1$ in the filter equations. Summing up the outputs of these neuron pairs will produce an output that is the same as if $\sigma(\cdot)$ was the linear function.

The output of the GNN for the two graphs takes the form

$$\boldsymbol{Y}_n = \boldsymbol{H}_n\left(\boldsymbol{S}\right)\boldsymbol{X} = \boldsymbol{u}_n \boldsymbol{u}_n^T \boldsymbol{X}, \quad n = 1, \ldots, N \tag{74}$$

$$\hat{\boldsymbol{Y}}_n = \boldsymbol{H}_n\left(\hat{\boldsymbol{S}}\right)\hat{\boldsymbol{X}} = \hat{\boldsymbol{u}}_n \hat{\boldsymbol{u}}_n^T \hat{\boldsymbol{X}}, \quad n = 1, \ldots, N \tag{75}$$

$$\boldsymbol{Y}_n = \boldsymbol{u}_n\left[\boldsymbol{u}_n^T \boldsymbol{x}_1, \ldots, \boldsymbol{u}_n^T \boldsymbol{x}_D\right], \quad n = 1, \ldots, N \tag{76}$$

$$\hat{\boldsymbol{Y}}_n = \hat{\boldsymbol{u}}_n\left[\hat{\boldsymbol{u}}_n^T \hat{\boldsymbol{x}}_1, \ldots, \hat{\boldsymbol{u}}_n^T \hat{\boldsymbol{x}}_D\right], \quad n = 1, \ldots, N \tag{77}$$

Now we assume that $\boldsymbol{X}^T \boldsymbol{u}_n \neq \boldsymbol{0}$ for all eigenvectors $\boldsymbol{u}_n$, $n = 1, \ldots, N$. As a result, there exist at least one column in each $\boldsymbol{Y}_n$ that is not equal to the zero column. We can then collect one nonzero column from each $\boldsymbol{Y}_n$ and form a matrix $\boldsymbol{M}$ as:

$$\boldsymbol{M} = \left[\boldsymbol{u}_1\left(\boldsymbol{u}_1^T \boldsymbol{x}_i\right), \ldots, \boldsymbol{u}_N\left(\boldsymbol{u}_N^T \boldsymbol{x}_j\right)\right] = \left[\boldsymbol{u}_1 \alpha_1, \ldots, \boldsymbol{u}_N \alpha_N\right] = \boldsymbol{U}\begin{bmatrix} \alpha_1, 0, \ldots, 0 \\ 0, \alpha_2, \ldots, 0 \\ \vdots \\ 0, 0, \ldots, \alpha_N \end{bmatrix} = \boldsymbol{U}\boldsymbol{A}, \tag{78}$$

where $\boldsymbol{x}_i$, $\boldsymbol{x}_j$ are columns of $\boldsymbol{X}$ such that $\boldsymbol{u}_1^T \boldsymbol{x}_i \neq 0$, $\boldsymbol{u}_N^T \boldsymbol{x}_j \neq 0$, $\boldsymbol{A}$ is a diagonal matrix and $\alpha_n \neq 0$ for all $n$. If we also collect the corresponding columns for each $\hat{\boldsymbol{Y}}_n$ we can form:

$$\hat{\boldsymbol{M}} = \left[\hat{\boldsymbol{u}}_1\left(\hat{\boldsymbol{u}}_1^T \hat{\boldsymbol{x}}_i\right), \ldots, \hat{\boldsymbol{u}}_N\left(\hat{\boldsymbol{u}}_N^T \hat{\boldsymbol{x}}_j\right)\right] = \left[\hat{\boldsymbol{u}}_1 \hat{\alpha}_1, \ldots, \hat{\boldsymbol{u}}_N \hat{\alpha}_N\right] = \hat{\boldsymbol{U}}\begin{bmatrix} \hat{\alpha}_1, 0, \ldots, 0 \\ 0, \hat{\alpha}_2, \ldots, 0 \\ \vdots \\ 0, 0, \ldots, \alpha_N \end{bmatrix} = \hat{\boldsymbol{U}}\hat{\boldsymbol{A}}, \tag{79}$$

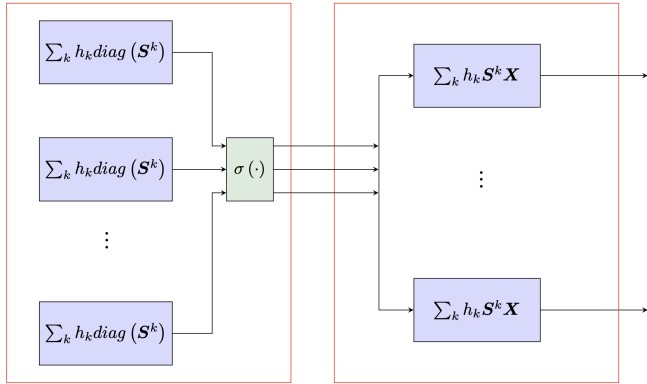

Figure 7: GNN architecture

where $\hat{A}$ is a diagonal matrix but $\hat{\alpha}_n$ are not necessarily nonzero, as $\hat{u}_1^T \hat{x}_i$, $\hat{u}_N^T$ are not necessarily nonzero. Since $\mathcal{G}$, $\hat{\mathcal{G}}$ are nonisomorphic and Assumption H.1 holds, we can use Lemma H.2. Consequently, there are no permutation matrix $\Pi$ and diagonal $\pm 1$ matrix $D$ such that $U = \Pi \hat{U} D$. This implies that there does not exist a permutation matrix $\Pi$ such that $M = \Pi \hat{M}$. To complete the proof, we consider the output of the considered GNN to be the concatenation of $Y_n$, $n = 1, \ldots, N$. In particular, the outputs for $\mathcal{G}$, $\hat{\mathcal{G}}$ are:

$$Y = [Y_1, Y_2, \ldots, Y_N] \tag{80}$$

$$\hat{Y} = \left[ \hat{Y}_1, \hat{Y}_2, \ldots, \hat{Y}_N \right]. \tag{81}$$

The columns of $M$, $\hat{M}$ are also columns of $Y$, $\hat{Y}$. Since there does not exist a permutation matrix $\Pi$ such that $M = \Pi \hat{M}$, there does not exist $\Pi$ such that $Y = \Pi \hat{Y}$ and the GNN decides that $\mathcal{G}$, $\hat{\mathcal{G}}$ are nonisomorphic. Note that the same analysis is applicable if we assume that $\hat{X}^T \hat{u}_n \neq 0$ for all eigenvectors $\hat{u}_n$, $n = 1, \ldots, N$ and is therefore omitted. Now our proof is complete. $\square$

We also extend the results of Theorems 4.1, 5.2, 6.1 to incorporate the cases where the eigenvalues of the two graphs are the same.

**Theorem H.4** *Let $\mathcal{G}$, $\hat{\mathcal{G}}$ be nonisomorphic graphs. Then there exists a GNN with modules as in Fig. 3 or as in Fig.4 that tells the two graphs apart if:*

1. *Assumption 2.1 holds or*

2. *Assumption H.1 holds and*

    (a) *There is no permutation matrix $\Pi$ such that $|U| = \Pi |\hat{U}|$, or*
    (b) *$|U^T| u_n \neq 0$ for all eigenvectors $u_n$ or $|\hat{U}^T| \hat{u}_n \neq 0$ for all eigenvectors $\hat{u}_n$.*

*Proof:* The proof of Case 1 can be found in Appendix F. In order to prove case 2 of Theorem H.4 we use the architecture illustrated in Fig. 7. This GNN is designed with 2 layers, each of them consisting of $N$ neurons. Recall from the previous proofs that there exists a graph filter $H(S)$ with unique parameters $h$ that isolates one eigenvalue (the $n$-th eigenvalue) and sets it to one, i.e.,

$$\tilde{H}(\lambda_i) = \begin{cases} 1, & \text{if } i = n \\ 0, & \text{if } i \neq k \end{cases} \tag{82}$$

Since the considered graphs have $N$ distinct eigenvalues, we can build the first layer of Fig. 7 with $N$ neurons described by the following set of equations:

$$y_n = \sigma \left( \text{diag} \left( H_n^{(1)}(S) \right) \right), \quad n = 1, \ldots, N \tag{83}$$

$$\tilde{H}_n^{(1)}(\lambda_i) = \begin{cases} 1, & \text{if } i = n \\ 0, & \text{if } i \neq n \end{cases}, \quad n = 1, \ldots, N \tag{84}$$

Under the above parametrization, the filter $\boldsymbol{H}_n^{(1)}(\boldsymbol{S})$ takes the form $\boldsymbol{H}_n^{(1)}(\boldsymbol{S}) = \boldsymbol{u}_n\boldsymbol{u}_n^T$, where $\boldsymbol{u}_n$ is the eigenvector corresponding to the $n-$th eigenvalue of $\boldsymbol{S}$. Then the output of the first layer for the two graphs takes the form:

$$\boldsymbol{y}_n = \sigma\left(\operatorname{diag}\left(\boldsymbol{u}_n\boldsymbol{u}_n^T\right)\right) = |\boldsymbol{u}_n|^2, \quad n = 1, \ldots, N \tag{85a}$$

$$\hat{\boldsymbol{y}}_n = \sigma\left(\operatorname{diag}\left(\hat{\boldsymbol{u}}_n\hat{\boldsymbol{u}}_n^T\right)\right) = |\hat{\boldsymbol{u}}_n|^2, \quad n = 1, \ldots, N \tag{85b}$$

Since both $\boldsymbol{S}$, $\hat{\boldsymbol{S}}$ have distinct eigenvalues, we can concatenate the output of each neuron and result in layer-1 outputs as:

$$\boldsymbol{Y}^{(1)} = |\boldsymbol{U}|, \quad \hat{\boldsymbol{Y}}^{(1)} = |\hat{\boldsymbol{U}}| \tag{86}$$

If there does not exist a permutation matrix $\boldsymbol{\Pi}$ such that $|\boldsymbol{U}| = \boldsymbol{\Pi}|\hat{\boldsymbol{U}}|$, one layer is sufficient and the proposed GNN can tell the two graphs apart.

For the second layer of the GNN in Fig. 7 we consider the following parametrization:

$$\boldsymbol{Y}_n = \left(\boldsymbol{H}_n^{(2)}(\boldsymbol{S})\boldsymbol{X}\right), \quad n = 1, \ldots, N \tag{87}$$

$$\tilde{\boldsymbol{H}}_n^{(2)}(\lambda_i) = \begin{cases} 1, & \text{if } i = n \\ 0, & \text{if } i \neq n \end{cases}, \quad n = 1, \ldots, N, \tag{88}$$

where $\boldsymbol{X} = \boldsymbol{Y}^{(1)} = |\boldsymbol{U}|$ is the output of the first layer. Then the final output of the GNN for the two graphs can be written as:

$$\boldsymbol{Y}_n = \boldsymbol{H}_n^{(2)}(\boldsymbol{S})\boldsymbol{X} = \boldsymbol{u}_n\boldsymbol{u}_n^T|\boldsymbol{U}|, \quad n = 1, \ldots, N \tag{89}$$

$$\hat{\boldsymbol{Y}}_n = \boldsymbol{H}_n^{(2)}\left(\hat{\boldsymbol{S}}\right)\hat{\boldsymbol{X}} = \hat{\boldsymbol{u}}_n\hat{\boldsymbol{u}}_n^T|\hat{\boldsymbol{U}}|, \quad n = 1, \ldots, N \tag{90}$$

$$\boldsymbol{Y}_n = \boldsymbol{u}_n\left[\boldsymbol{u}_n^T|\boldsymbol{u}_1|, \ldots, \boldsymbol{u}_n^T|\boldsymbol{u}_N|\right], \quad n = 1, \ldots, N \tag{91}$$

$$\hat{\boldsymbol{Y}}_n = \hat{\boldsymbol{u}}_n\left[\hat{\boldsymbol{u}}_n^T|\hat{\boldsymbol{u}}_1|, \ldots, \hat{\boldsymbol{u}}_n^T|\hat{\boldsymbol{u}}_N|\right], \quad n = 1, \ldots, N \tag{92}$$

If we assume that $|\boldsymbol{U}^T|\boldsymbol{u}_n \neq \boldsymbol{0}$ for all eigenvectors $\boldsymbol{u}_n$, or $|\hat{\boldsymbol{U}}^T|\hat{\boldsymbol{u}}_n \neq \boldsymbol{0}$ for all eigenvectors $\hat{\boldsymbol{u}}_n$, we can use the same steps as in the proof of Theorem H.3 and show that the proposed GNN decides that the two graphs are different. Note that in layer 1 we can use the stochastic modules in Fig. 3a and the proof still holds, since the filter with parameters as in (84) yields:

$$\operatorname{cov}\left[\boldsymbol{z}; \boldsymbol{S}\right] = \sum_{k=0}^{K-1} h_k\boldsymbol{S}^k \sum_{m=0}^{K-1} h_m\boldsymbol{S}^m = \boldsymbol{u}_n\boldsymbol{u}_n^T\boldsymbol{u}_n\boldsymbol{u}_n^T = \boldsymbol{u}_n\boldsymbol{u}_n^T, \tag{93}$$

and the same output as in (86) can be produced. Also, by using Proposition 6.1 we can substitute the modules in the first layer with the modules in Fig. 4b and the proof still holds. $\square$

## H.2 GRAPHS WITH THE SAME EIGENVALUES WHICH ARE NOT DISTINCT.

The last case appears when the graph adjacencies have the same eigenvalues, which are not distinct and have the same multiplicities. This case is more complicated, since the two graphs can be nonisomorphic even if there exist a permutation matrix $\boldsymbol{\Pi}$ and a diagonal matrix $\boldsymbol{D}$ such that $\boldsymbol{U} = \boldsymbol{\Pi}\hat{\boldsymbol{U}}\boldsymbol{D}$ (the condition in Lemma H.2 does not hold). Analysis and results for this case are left for future work.

## I GNNS AND ISOMORPHIC GRAPHS

The core of this paper studies the ability of GNNs to distinguish between nonisomorphic graphs. Another important question is whether a GNN can tell if two graphs are isomorphic. The answer is affirmative. GNNs are permutation equivariant architectures and can always detect isomorphic graphs. To make things concrete, we present the following proposition:

**Proposition I.1** . *Let $\mathcal{G}$, $\hat{\mathcal{G}}$ be two isomorphic graphs, i.e., $\boldsymbol{S} = \boldsymbol{\Pi}\hat{\boldsymbol{S}}\boldsymbol{\Pi}^T$. Also let $\boldsymbol{X}$, $\hat{\boldsymbol{X}}$ be the graph signals associated with $\mathcal{G}$, $\hat{\mathcal{G}}$ that satisfy $\boldsymbol{X} = \boldsymbol{\Pi}\hat{\boldsymbol{X}}$. Then any GNN with modules as in* (1) *decides the two graphs are the same.*

*Proof:* To prove this proposition, it suffices to show that the output $\boldsymbol{Y}$ in (1) is permutation equivariant. To see this, consider the graph adjacencies $\boldsymbol{S}$ and $\hat{\boldsymbol{S}}$ such that $\hat{\boldsymbol{S}} = \boldsymbol{\Pi}\boldsymbol{S}\boldsymbol{\Pi}^T$, where $\boldsymbol{\Pi}$ is a permutation matrix. Then equation (1) gives:

$$\hat{\boldsymbol{Y}} = \sigma\left(\sum_{k=0}^{K-1} \hat{\boldsymbol{S}}^k \hat{\boldsymbol{X}}\boldsymbol{H}_k\right) \stackrel{(1)}{=} \sigma\left(\sum_{k=0}^{K-1} h_k\left(\boldsymbol{\Pi}\boldsymbol{S}^k\boldsymbol{\Pi}^T\right)\boldsymbol{\Pi}\boldsymbol{X}\boldsymbol{H}_k\right) \stackrel{(2)}{=} \sigma\left(\sum_{k=0}^{K-1} h_k\boldsymbol{\Pi}\boldsymbol{S}^k\boldsymbol{X}\boldsymbol{H}_k\right) \quad (94)$$

$$= \sigma\left(\boldsymbol{\Pi}\sum_{k=0}^{K-1} h_k\boldsymbol{S}^k\boldsymbol{X}\boldsymbol{H}_k\right) = \boldsymbol{\Pi}\boldsymbol{Y}, \quad (95)$$

where equality (1) holds because $\left(\boldsymbol{\Pi}\boldsymbol{S}\boldsymbol{\Pi}^T\right)^k = \boldsymbol{\Pi}\boldsymbol{S}^k\boldsymbol{\Pi}^T$ and equality (2) comes from the fact that $\boldsymbol{\Pi}^T\boldsymbol{\Pi} = \boldsymbol{I}$. Therefore, $\boldsymbol{Y}$ is equivariant in permutation. Overall GNNs with modules as in (1) produce permutation equavariant outputs for isomorphic graphs.

**Proposition I.2** . *Let $\mathcal{G}$, $\hat{\mathcal{G}}$ be two isomorphic graphs. Then any GNN with modules as in Fig. 3 or Fig. 4 decides that the two graphs are the same.*

*Proof:* To prove this proposition, it suffices to show that the output in (10) is permutation equivariant. To see this, consider two graph adjacencies $\boldsymbol{S}$ and $\hat{\boldsymbol{S}}$ such that $\hat{\boldsymbol{S}} = \boldsymbol{\Pi}\boldsymbol{S}\boldsymbol{\Pi}^T$, where $\boldsymbol{\Pi}$ is a permutation matrix. Then Equation (10) gives:

$$\hat{\boldsymbol{y}} = \sigma\left(\sum_{k=0}^{K-1} h_k\mathrm{diag}\left(\hat{\boldsymbol{S}}^k\right)\right) \stackrel{(1)}{=} \sigma\left(\sum_{k=0}^{K-1} h_k\mathrm{diag}\left(\boldsymbol{\Pi}\boldsymbol{S}^k\boldsymbol{\Pi}^T\right)\right) \stackrel{(2)}{=} \sigma\left(\sum_{k=0}^{K-1} h_k\boldsymbol{\Pi}\mathrm{diag}\left(\boldsymbol{S}^k\right)\right) \quad (96)$$

$$= \sigma\left(\boldsymbol{\Pi}\sum_{k=0}^{K-1} h_k\mathrm{diag}\left(\boldsymbol{S}^k\right)\right) = \boldsymbol{\Pi}\boldsymbol{y}, \quad (97)$$

where equality (1) holds because $\left(\boldsymbol{\Pi}\boldsymbol{S}\boldsymbol{\Pi}^T\right)^k = \boldsymbol{\Pi}\boldsymbol{S}^k\boldsymbol{\Pi}^T$ and equality (2) comes from the fact that $\mathrm{diag}\left(\boldsymbol{\Pi}\boldsymbol{S}\boldsymbol{\Pi}^T\right) = \boldsymbol{\Pi}\mathrm{diag}\left(\boldsymbol{S}\right)$. The output $\boldsymbol{y}$ is permutation equivariant and we can conclude that the proposed architectures produce permutation equivariant outputs for isomorphic graphs.

## J   GNNS VS SPECTRAL DECOMPOSITION

In this paper, we discuss the ability of GNNs to distinguish between different graphs. Our analysis uses spectral decomposition tools and provides conditions under which a GNN can tell two graphs apart. These conditions are related to the eigenvalues and eigenvectors of the graph operators. Therefore, it is natural to study the similarities and differences of GNNs and spectral decomposition algorithms.

### J.1   THE TWO GRAPHS HAVE DIFFERENT EIGENVALUES

As explained in the main part of the paper, there always exists a GNN that can distinguish between a pair of graphs with different eigenvalues. Furthermore, computing the eigenvalues of the two graphs can also attest that the two graphs are nonisomorphic. Therefore, the two approaches are equally powerful. The difference lies in the fact that a GNN needs to be trained to perform the isomorphism test, whereas the spectral decomposition is unsupervised. On the other hand, computing the spectral decomposition for real graphs can be computationally very challenging.

### J.2   THE TWO GRAPHS HAVE THE SAME SET OF EIGENVALUES THAT ARE DISTINCT

This case is a bit more complicated. Since the eigenvalues are the same, one must resort to the eigenvectors to distinguish between the graphs. When the eigenvalues are distinct, the eigenvectors

of the graph are unique up to a sign for each eigenvector. To be more precise, let $\mathcal{G}$, $\hat{\mathcal{G}}$ be isomorphic graphs with eigenvectors $\boldsymbol{U}$, $\hat{\boldsymbol{U}}$ respectively. Then we have the following:

$$\boldsymbol{U} = \boldsymbol{\Pi}\hat{\boldsymbol{U}}\boldsymbol{D}, \tag{98}$$

where $\boldsymbol{\Pi}$ is a permutation matrix and $\boldsymbol{D}$ is a diagonal matrix with elements $\pm 1$. We observe the following:

**Remark J.1** *When Assumption H.1 holds, the eigenvectors of isomorphic graphs are not permutation equivariant, since there exists a sign ambiguity for each eigenvector. On the contrary, the produced GNN node embeddings are always permutation equivariant, according to Propositions I.2 and I.1. In other words, GNNs always produce equivariant node embeddings for isomorphic graphs, which is not the case for the spectral decomposition.*

If $\mathcal{G}$, $\hat{\mathcal{G}}$ are nonisomorphic the story is different. According to Lemma H.2, there does not exist permutation matrix $\boldsymbol{\Pi}$ such that $\boldsymbol{U} = \boldsymbol{\Pi}\hat{\boldsymbol{U}}\boldsymbol{D}$ and the GNNs detect nonisomorphic graphs under Theorem H.3, or Theorem H.4. Let us focus on the conditions of Theorem H.4 i.e.,

(a) There does not exist permutation matrix $\boldsymbol{\Pi}$ such that $|\boldsymbol{U}| = \boldsymbol{\Pi}|\hat{\boldsymbol{U}}|$,

(b) $|\boldsymbol{U}^T|\boldsymbol{u}_n \neq \boldsymbol{0}$ for all eigenvectors $\boldsymbol{u}_n$ or $|\hat{\boldsymbol{U}}^T|\hat{\boldsymbol{u}}_n \neq \boldsymbol{0}$ for all eigenvectors $\hat{\boldsymbol{u}}_n$.

We see that these conditions involve the eigenvectors of the graphs and therefore we can construct an eigen-based algorithm with the same guarantees. Note that these guarantees are only sufficient and there might be cases where the GNNs can distinguish between nonisomorphic graphs, whereas an algorithm based on the above conditions might fail. Furthermore, calculating the complete set of eigenvectors of a real graph might be computationally prohibitive.

## J.3    THE TWO GRAPHS HAVE THE SAME MULTISET OF EIGENVALUES THAT ARE NOT DISTINCT

**Scenario 1: The graphs are isomorphic.** GNNs will always produce equivariant embeddings for isomorphic graphs. On the contrary, eigenvectors are not unique and they will not provide equivariant representations (up to scaling) for isomorphic graphs.

**Scenario 2: The graphs are nonisomorphic.** The GNN analysis for this case is relegated for future work. Regarding the spectral decomposition, we need to resort to eigenvectors, which are not unique. Therefore, detecting nonisomorphic graphs is challenging.

## J.4    STABILITY AND DISCRIMINABILITY OF GNNS

From our discussion so far, we have observed similarities and differences between the functionality of GNNs and the spectral decomposition of the graph. There is one more fundamental difference that has not yet been discussed and involves the stability and discriminability properties of GNNs (Gama et al., 2020). In particular, a GNN is stable under small perturbations of the graph operator, i.e., the output of a GNN is similar for 'similar' graphs. On the other hand, small perturbations of the graph can result in essential changes in the eigenvalues and eigenvectors of the graph operator, which makes the spectral decomposition more unstable. Therefore, there seems to be a stability vs. discriminability trade-off between GNNs and spectral decomposition. However, the architectural nonlinearities allow GNNs to be both stable and discriminative.

To recap, the conditions of this paper involve the eigenvalues and eigenvectors of the graph operator. Compared to eigen-based algorithms, there is an advantage of GNNs when the eigenvalues are exactly the same with the same multiplicities. This is due to the fact that the eigenvectors of a graph operator are not unique and therefore isomorphic graphs do not admit permutation equivariant eigenvectors, whereas GNNs always produce permutation equivariant node embeddings for isomorphic graphs. On the other hand, when the eigenvalues are different, GNNs and spectral decomposition are equally powerful. Furthermore, GNNs are robust to small changes of the graph, which is not the case for spectral decomposition. Finally, the spectral decomposition is computationally heavy and unsupervised, but GNNs are lighter to execute and require training.

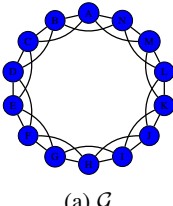

(a) $\mathcal{G}$

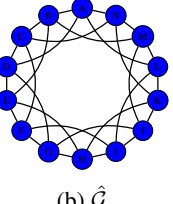

(b) $\hat{\mathcal{G}}$

Figure 8: CSL graphs

## K EXPERIMENTAL DETAILS

In this appendix we provide further details on the experiments presented in the main paper.

### K.1 EXPERIMENTS ASSOCIATED WITH THE GRAPHS IN FIGS. 1 AND 2

In Tables 8, 9 we present the eigenvalues and the sum of the corresponding eigenvectors of the graphs in Figs. 1, 2 respectively.

Table 8: Eigenvalue and eigenvector information for the graphs in Fig. 1.

| GRAPH | | 1 | 2 | $n$ 3 | 4 | 5 | 6 |
|---|---|---|---|---|---|---|---|
| $\mathcal{G}$ | $\lambda_n$ | 3 | 1 | -2 | -2 | 0 | 0 |
| | $\boldsymbol{u}_n^T\mathbf{1}$ | -2.45 | 0 | 0 | 0 | 0 | 0 |
| $\hat{\mathcal{G}}$ | $\hat{\lambda}_n$ | 3 | -3 | 0 | 0 | 0 | 0 |
| | $\hat{\boldsymbol{u}}_n\mathbf{1}$ | -2.45 | 0 | 0 | 0 | 0 | 0 |

Table 9: Eigenvalue and eigenvector information for the graphs in Fig. 2.

| GRAPH | | 1 | 2 | 3 | 4 | 5 | $n$ 6 | 7 | 8 | 9 | 10 |
|---|---|---|---|---|---|---|---|---|---|---|---|
| $\mathcal{G}$ | $\lambda_n$ | 2.303 | 1.618 | 1.303 | 1 | 0.618 | -2.303 | -1.618 | -0.618 | -1 | -1.303 |
| | $\boldsymbol{u}_n^T\mathbf{1}$ | 3.048 | 0 | 0 | -0.816 | 0 | 0 | 0 | 0 | 0 | -0.210 |
| $\hat{\mathcal{G}}$ | $\hat{\lambda}_n$ | 2.303 | 1.861 | 1 | 0.618 | 0.618 | 0.254 | -1.303 | -1.618 | -1.618 | -2.115 |
| | $\hat{\boldsymbol{u}}_n\mathbf{1}$ | 3.048 | 0 | -0.816 | 0 | 0 | 0 | -0.210 | | 0 | 0 |

We observe that $\mathcal{G}$ and $\hat{\mathcal{G}}$ in both figures admit a different set of eigenvalues. However, the eigenvectors that correspond to the eigenvalues that differentiate them are orthogonal to the vector of all-ones (they sum up to zero). Therefore, the WL algorithm and GNNs with $\boldsymbol{x} = \mathbf{1}$ input fail to tell them apart.

### K.2 DETAILS ON THE EXPERIMENTS OF SECTION 7

In Fig. 8 we present a paradigm of two graphs in the CSL dataset that belong to different classes. It is clear from the figure that the two graphs consist of nodes that all have degrees equal to $4$. Therefore, $\boldsymbol{x} = \mathbf{1}$ is an eigenvector of both graphs and orthogonal to the remaining eigenvectors. Any valuable information that separates the two graphs is lost when we run the WL algorithm or feed a GNN with $\boldsymbol{x} = \mathbf{1}$.

Next, we present the details on the experiments of section 7.2. For the most part, we use the specifications suggested in (Xu et al., 2019). In particular, we train a 4-layer graph neural network where the output of each layer and the input are passed through a graph pooling layer and then a

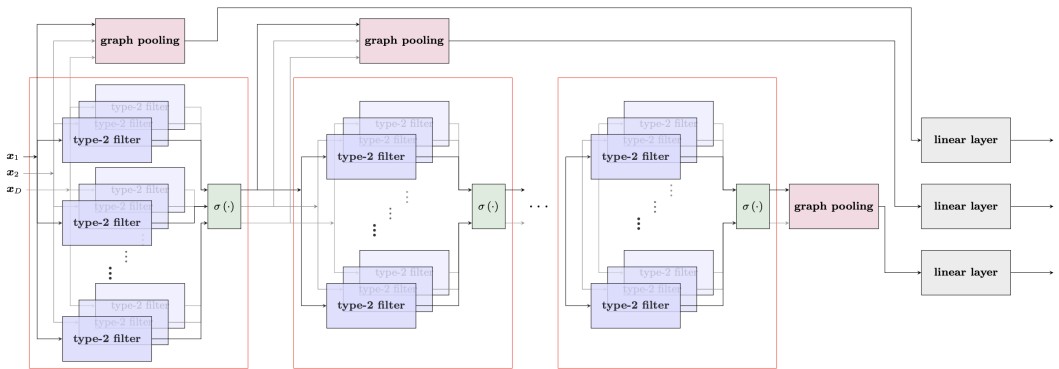

Figure 9: GNN architecture

linear classifier. A schematic representation of the considered architecture is presented in Fig. 9. The nonlinearity used in our experiments is the ReLU and the readout function that performs graph pooling is $\mathbf{1}^T \boldsymbol{X}^{(l)}$ for $l = 0, \ldots, 5$. $\boldsymbol{X}^{(l)}$ represents the output of layer $l$ with $\boldsymbol{X}^{(0)} = \boldsymbol{X}$. For each type-2 GNN block, we only use 1 tap for $k = 1$. This is due to the fact that we pass the output of every layer to the final classifier, so additional taps might be redundant.

To train the proposed architecture, we use Adam optimizer with a learning rate equal to $10^{-2}$, batch size equal to $128$ and a dropout ratio equal to $0.5$. Training is carried out over $200$ epochs with $50$ iterations per epoch. To assess the performance of the proposed architecture, we divide each dataset into $50 - 50$ training-testing splits and apply 10-fold cross-validation. The only parameter we tune is the hidden dimension for each layer. In particular, the number of modules for each layer is the same and we tune over $\{8, 16, 32, 64, 128, 256\}$ modules.

We also compare our proposed architecture with GIN (Xu et al., 2019) initialized with $x = 1$ and GIN initialized according to equation (14). We use the publicly available code[2] provided by the authors. We use the exact same specification for fair comparisons and tune the hidden layer over $\{8, 16, 32, 64, 128, 256\}$ dimensions.

All experiments are conducted on a Linux server with NVIDIA RTX 3080 GPU. The data[13] are publicly available, and the code of the proposed architectures with all the experiments can be found in this repository[4].

---

[2]https://github.com/weihua916/powerful-gnns
[3]https://pytorch-geometric.readthedocs.io/en/latest/
[4]https://github.com/tempcode100/gnns-are-powerful

