# OpenReview forum: "Graph Neural Networks Are More Powerful Than we Think"
_ICLR.cc/2023/Conference — Submitted to ICLR 2023_

### Official Review · Reviewer_jQZJ · 2022-10-24

**Confidence:** 2
**Correctness:** 3
**Technical Novelty And Significance:** 2
**Empirical Novelty And Significance:** 1
**Recommendation:** 3

**Clarity, Quality, Novelty And Reproducibility:**

Clarity

I think there is room for improvement in the architecture description and mathematical claims.
In several theorems (Theorem 4.1, Proposition 4.1, Proposition 6.1, Corollary 6.2), the architecture is explained only schematically using figures. For example, there is no explanation in the main text corresponding to the type-1 architecture in Figure 4. While figures enable us to understand concepts smoothly, they are auxiliary tools and unsuitable for accurate discussions. The architecture should be explained in the text to exclude ambiguity.

I would suggest writing about how a GNN determines whether two graphs are isomorphic based on the output signals obtained from an input pair of graphs (e.g., in Theorem 2.2). Judging from the proof, the GNN decides two graphs are isomorphic if and only if there exists a permutation matrix $\Pi$ such that $Y'=\Pi Y$, where $Y$ and $Y'$ are the outputs of input graphs $G$ and $G'$, respectively (this is efficiently computable by sorting the value of $Y$ and $Y'$)

Theorem 4.1 and 6.1 claimed that two GNNs are "equivalent" but did not define "equivalent" mathematically. Therefore I would suggest writing its formal definition.

(minor comments) Contribution (C2) claimed that "the WL algorithm is a limit [...] only when we use the all-one vectors as an input". Strictly speaking, this is not strictly true. It is true that the WL algorithm is upper bound when we use the all-one vector. However, it is not shown that the converse holds, i.e., there is the possibility that inputs are not all-one and the expressive power is lower than the WL algorithm.


Quality
Contribution (C3) claimed that GNNs have a higher discriminative ability when "node features are initialized with random white noise." However, I think this is a slightly confusing expression. In my understanding, white random noise input means that we draw a sample from a Gaussian distribution and feed the sample to the GNN. However, the GNN in Figure 3(a) takes the expectation of the input variable. This operation is impossible from a single realization of a random variable $x$. It requires the information of the random variable $x$ (e.g., distribution of $x$).

Experiment results (Table 4) showed that the proposed method was not superior to the baseline when all-one vectors were used. Since this study has a theoretical nature, I do not think this result does not solely incur the significance of this study significantly. On the other hand, I believe we can obtain many implications from this result. For example, there is a gap between the theoretical graph homomorphism identification capability and the practical prediction accuracy on graphs. I would suggest discussing what we can learn from the numerical evaluations.


Novelty

Several studies theoretically and experimentally enhanced the representational capability of GNNs by enriching the graph input (e.g., [You et al., 21], [Ishiguro et al., 20] and reference therein). Moreover, some of them have theoretically shown that GNNs they proposed have better discriminative power than the 1-WL algorithm in the graph isomorphism problem. Therefore, I think the novelty of this study concerning the strategy of improvement is limited. On the other hand, focusing on graph spectra is an good approach because, as far as I know, few studies pay attention to it. In addition, Assumption 2.1 for the identification is general in the sense that many graph pairs satisfy this condition.

[You et al., 21] https://ojs.aaai.org/index.php/AAAI/article/view/17283
[Ishiguro et al., 20] https://arxiv.org/abs/2006.06909


Reproducibility

This paper provides the references and specifications of the datasets. Also, the code URL and hyperparaemters are available in the appendix. Therefore, although there is no guarantee of perfect reproduction, I think we can implement the code to reproduce the experiments to some degree.


Questions

Looking at theoretical analyses (e.g., Theorem 4.1, 5.2, and 6.1), I am wondering whether they imply that we can increase the discriminative power of the WL algorithm by adopting the numbers of cycles of length $k$ ($1\leq k\leq K$) for an appropriate $K$ as the initial colors?

**Details Of Ethics Concerns:**

N.A.

**Strength And Weaknesses:**

Strengths
- Sufficient conditions (Assumption 2.1) are general and have broad applicability.

Weaknesses
- Mathematical arguments are somewhat informal, and the description has room for improvement.
- Experiments on real data have not shown that the proposed methods (in terms of both architectures and input features) are practically effective on prediction tasks on graphs.

**Summary Of The Paper:**

GNNs are known to have a discriminative power of at most 1-WL algorithm in the graph isomorphism problem. This paper pointed out that this result assumed that inputs are the all-one vectors and analyzed the expressive power of GNNs in the graph isomorphism problem when inputs are not necessarily all-one.
First, this paper gave sufficient conditions on the spectrum of the adjacency matrices and the input vectors for a GNN to discriminate between two non-isomorphic graphs (Theorem 2.2). In particular, it was shown that this condition held when inputs were an appropriate number of Gaussian random noise (Proposition 4.1). Also, the condition is achieved when inputs are numbers of cycles of appropriate length in the graphs (Theorem 5.1. Theorem 6.1).

**Summary Of The Review:**

The sufficient conditions derived in this paper for a GNN to discriminate a pair of non-isomorphic graphs are flexible. On the other hand, enriching the input of GNNs may not be a novel approach in the theoretical analysis of the expressive power of GNNs. In addition, the statements and architectural descriptions are somewhat informal, and there is room for improvement.

---

> ### Comment · Reviewer_jQZJ · 2022-11-17
> **Post Discussion Period Comment**
>
> So far the authors did not have any comment. I want to keep my score

---

> > ### Author Response · Authors · 2022-11-17
> > **Response to reviewer's comment**
> >
> > We thank the reviewer for their feedback. We are preparing our response which will be submitted before the deadline on November 18.

---

> > > ### Comment · Reviewer_jQZJ · 2022-11-17
> > > **I have mistakenly recognized the reviewing process.**
> > >
> > > I mistakenly understood the reviewing process and thought that I should determine the final score now.I understand it is not now and that the current period is the discussion phase between authors and reviewers.
> > >
> > > I am sorry for the confusion. I am looking forward to the authors' responses.

---

> > > > ### Author Response · Authors · 2022-11-17
> > > > **Clarification about discussion period**
> > > >
> > > > Thank you for acknowledging this. Note that there is Discussion Stage 2 from November 18 to December 12,
> > > > where authors, reviewers and AC will have the chance to discuss.
> > > >
> > > > We are looking forward to a fruitful discussion.

---

> ### Author Response · Authors · 2022-11-18
> **Response to reviewer's comments (2/2)**
>
> > Experiments on real data have not shown that the proposed methods (in terms of both architectures and input features) are practically effective on prediction tasks on graphs.
>
> > Experiment results (Table 4) showed that the proposed method was not superior to the baseline when all-one vectors were used. Since this study has a theoretical nature, I do not think this result does not solely incur the significance of this study significantly. On the other hand, I believe we can obtain many implications from this result. For example, there is a gap between the theoretical graph homomorphism identification capability and the practical prediction accuracy on graphs. I would suggest discussing what we can learn from the numerical evaluations.
>
> See our general comment. The experiments aim to study the effect of using anonymous all-one inputs vs anonymous random inputs (which translate to adjacency power diagonals) on the expressivity of GNNs. We study the plain-vanilla convolutional GNN model and the GIN model, which are standard architectures, that are falsely considered to be less expressive than the WL-test. We see significant benefit in 3 datasets (CSL, REDDITBINARY, REDDITMULTI) and for the remaining datasets the performance is statistically the same. This is expected since the information loss due to the all-one vector is insignificant for the graph classification task in these datasets. In general, our results show that representational capability is important for this prediction task. However, what is equally important is generalization capability (which we do not study), as well as training data and optimization (which we do not control). This discussion is included in the revised manuscript.
>
> > Several studies theoretically and experimentally enhanced the representational capability of GNNs by enriching the graph input (e.g., [You et al., 21], [Ishiguro et al., 20] and reference therein). Moreover, some of them have theoretically shown that GNNs they proposed have better discriminative power than the 1-WL algorithm in the graph isomorphism problem. Therefore, I think the novelty of this study concerning the strategy of improvement is limited.
>
> We thank the reviewer for bringing [You et al., 21], [Ishiguro et al., 20] to our attention, which we cite in the revised version. We refer the reviewer to the discussion in the general comment. To reemphasize, we claim that a GNN is powerful if it can generate discriminative outputs for inputs that are not discriminative. We prove that a GNN can do this for all graphs with different eigenvalues. Interestingly measuring the variance of the stochastic output is equivalent to computing features associated with the diagonals of the graph adjacency powers. Therefore, our contribution is not just on enriching the GNN input to increase the representational capability of GNNs.
>
> Contrary to our work, [You et al., 21], use a different algorithm to produce discriminative inputs, and [Ishiguro et al., 20] use the available attribute node information. Of course, there is nothing wrong with it, and Theorem 2.2 can be used to describe the expressive power of both. However, in both cases and also in the majority of the literature, this increased discriminability comes from the handcrafted inputs, and conclusions cannot be drawn for the general representational capabilities of GNNs, as explained in the general comment.
>
> > Looking at theoretical analyses (e.g., Theorem 4.1, 5.2, and 6.1), I am wondering whether they imply that we can increase the discriminative power of the WL algorithm by adopting the numbers of cycles of length $k$ $(1\leq k\leq K)$ for an appropriate $K$ as the initial colors?
>
> The reviewer is right; the number of k-node cycles is included in the diagonal of the k-th power of the adjacency. As we mention in the Appendix, when summation is a proper hash function, WL is equivalent to certain GNNs and our Theorems can be applied to WL algorithm too. Furthermore, augmenting this information with the WL algorithm will increase the algorithms' ability to discriminate, as the WL cannot count cycles.

---

> > ### Comment · Reviewer_jQZJ · 2022-11-18
> > **Thank you for your response**
> >
> > I thank the authors for giving me reply to my review comments. I will carefully read their comments and update my review comments accordingly.

---

> > ### Comment · Reviewer_jQZJ · 2022-12-02
> > **Response**
> >
> > I thank the authors for responding to my comments.
> >
> > **About mathematical explanations**
> >
> > I understand that the equations corresponding to figures are certainly in the main text. My point was that the connection between figures and equations needed to be more explicit. Readers need to guess which equations each figure refers to. For example, I expected an explanation like "Figure 4 represents the schematic view of the architecture defined by (8)". So in this sense, the updated manuscript (e.g., Proposition 6.1) is OK.
> > Another thing is that the architectures presented in Figure 5 need precise definitions.
> >
> > **Criteria of isomorphism by GNNs**
> >
> > OK
> >
> > **About Contribution (C2)**
> >
> > OK
> >
> > **About Random Input**
> >
> > I understand that for Contribution (C3), input features are a random variable, not its realization. Since feeding a random variable as an input is not practical, we need to explore an equivalent deterministic model.
> >
> > **About Experiment Results**
> >
> > > We see significant benefit in 3 datasets (CSL, REDDITBINARY, REDDITMULTI) and for the remaining datasets the performance is statistically the same
> > > This is expected since the information loss due to the all-one vector is insignificant for the graph classification task in these datasets
> >
> > In my understanding, the authors claimed that in some datasets, compared with the vanilla GIN, the proposed input enhanced the prediction performance. I wonder how the authors judge that "the information loss due to the all-one vector is insignificant" in these datasets. If it would be based on the result presented in Table 4, there is a possibility that the argument might be a tautological argument.
> >
> > **Relation to prior work**
> >
> > I understand that this work is different from prior work because it showed that GNNs are powerful in the sense that they can discriminate graphs from "uninformative" inputs. However, I still have whether this claim is appropriate because it is practically impossible to feed a random variable as an input to GNNs. Since most existing works did not consider such situations, I wonder if this is a fair comparison.
> >
> > **Improvement of WL algorithm**
> >
> > OK
> >
> >
> > Additional comments
> >
> > The semantics of $\sigma$ is different between sections 4 and 5. Specifically, $\sigma$ represents the covariance operator in section 4, while it is a general activation function in section 5. So, I suggest reconsidering the notation.

---

> > > ### Author Response · Authors · 2022-12-05
> > > **Response to reviewer's new comments**
> > >
> > > > About Experiment Results
> > > In my understanding, the authors claimed that in some datasets, compared with the vanilla GIN, the proposed input enhanced the prediction performance. I wonder how the authors judge that "the information loss due to the all-one vector is insignificant" in these datasets. If it would be based on the result presented in Table 4, there is a possibility that the argument might be a tautological argument.
> > >
> > > Let us clarify that our argument "the information loss ..." refers to datasets that the two approaches work statistically the same. In the CSL dataset, where we see the most emphatic improvement, the graphs are regular and the all-one input omits any distinctive information between the graphs. The datasets, where performance is similar, also include graphs with eigenvectors that are orthogonal to one. These eigenvectors are usually associated with eigenvalues of low energy. As a result, omitting them may have a negligible effect. Having said that, one should take into account other deep learning components that affect the GNN performance, such as optimization, data distribution and training-testing splits, to name a few. Therefore, our observation should be viewed as an interpretation of the results and not as the only reason why the two approaches are similar.
> > >
> > > > About Random Input
> > > I understand that for Contribution (C3), input features are a random variable, not its realization. Since feeding a random variable as an input is not practical, we need to explore an equivalent deterministic model.
> > > > Relation to prior work
> > > I understand that this work is different from prior work because it showed that GNNs are powerful in the sense that they can discriminate graphs from "uninformative" inputs. However, I still wonder whether this claim is appropriate because it is practically impossible to feed a random variable as an input to GNNs.
> > >
> > > We kindly disagree with the reviewer. Our theoretical findings can be directly applied in practice. From a theoretical perspective feeding a GNN with random **uninformative input** can produce powerful outputs. From a practical perspective, we show that feeding the GNN with random input is doable, since the whole process has a closed-form deterministic representation (eq. (9)). Note that eq. (9) is also a proper GNN module that produces powerful representations **without any input**. Furthermore, as we mentioned in our previous response and explained in Appendix D, one can draw realizations of a white vector and feed the GNN.
> > >
> > > > Relation to prior work
> > > Since most existing works did not consider such situations, I wonder if this is a fair comparison.
> > >
> > > Existing works have claimed that the WL is the limit of the GNNs. We explain that this is the limit of GNNs with all-one inputs, not GNNs in general. GNNs with white input or GNNs with no input can separate graphs that fail the WL test.
> > > Existing works have also used random features and showed increased expressiveness compared to the WL. However, this improvement comes at the cost of permutation equivariance, which is a fundamental property. On the contrary, we show increased GNN expressiveness, while maintaining permutation equivariance.
> > >
> > >
> > > > Additional comments
> > > The semantics of $\sigma$ is different between sections 4 and 5. Specifically,  represents the covariance operator in section 4, while it is a general activation function in section 5. So, I suggest reconsidering the notation.
> > >
> > > We agree and thank the reviewer for this suggestion. In the updated manuscript $\sigma$ is the point-wise activation function.

---

> > > > ### Comment · Reviewer_jQZJ · 2022-12-12
> > > > **Response**
> > > >
> > > > I thank the authors for responsing my comments. It deepens my understanding about the paper.
> > > >
> > > > **About Random Input**
> > > >
> > > > I understand that although the stochastic GNN model Figure 3a is not practical as it is, we can transform it into the computatble model Figure 3b.
> > > >
> > > > **About Experiment Results**
> > > >
> > > > I understand that what the authors intended that "the information loss due to the all-one vector," is the interpretation of the experiment results about the graph topology and the input vectors.
> > > >
> > > > **Relation to prior work**
> > > >
> > > > > Existing works have also used random features and showed increased expressiveness compared to the WL.
> > > >
> > > > I think the existing works and this paper differ in using random variables. The existing works draw sample(s) from a random variable and feed them to a GNN (similar to the empirical variance model in Appendix D.) On the other hand, the stochastic GNN in Figure 3a feeds the random variable itself to a GNN. That is why I thought we could not compare them directly.

---

> > > > > ### Author Response · Authors · 2022-12-12
> > > > > **Response to reviewer**
> > > > >
> > > > > We are glad to see that the reviewer understands and accepts the significance of our contribution. We thank them for the discussion.

---

> ### Author Response · Authors · 2022-11-18
> **Response to reviewer's comments (1/2)**
>
> We would like to thank the reviewer for their detailed comments and valuable feedback.
>
> > Mathematical arguments are somewhat informal, and the description has room for improvement.
>
> > I think there is room for improvement in the architecture description and mathematical claims. In several theorems (Theorem 4.1, Proposition 4.1, Proposition 6.1, Corollary 6.2), the architecture is explained only schematically using figures.
>
> > For example, there is no explanation in the main text corresponding to the type-1 architecture in Figure 4. While figures enable us to understand concepts smoothly, they are auxiliary tools and unsuitable for accurate discussions. The architecture should be explained in the text to exclude ambiguity.
>
> We respectfully disagree with the reviewer. Our paper includes equations, theorems, proofs, and figures. Furthermore, the figures that the reviewer mentions contain the appropriate equations. To be more precise, the architecture in Theorem 4.1 and Proposition 4.1 are completely defined by eq. (8) and (10). The architectures in Proposition 6.1 are defined by Equations (2) and (11). The architectures in Corollary 6.2 are defined by Equations (2), (11) and (15). Type-1 architecture in Figure 4 is defined by Equation (11). All equation numbers are with respect to the original manuscript. However, since the reviewer raised this concern, we update the revised manuscript and provide more clear definitions.
>
>
> > I would suggest writing about how a GNN determines whether two graphs are isomorphic based on the output signals obtained from an input pair of graphs (e.g., in Theorem 2.2). Judging from the proof, the GNN decides two graphs are isomorphic if and only if there exists a permutation matrix $\mathbf{\Pi}$ such that $\mathbf{Y}^{\prime}=\mathbf{\Pi}\mathbf{Y}$, where $\mathbf{Y}$ and $\mathbf{Y}^{\prime}$ are the outputs of input graphs $G$ and $G^{\prime}$, respectively (this is efficiently computable by sorting the value of $\mathbf{Y}$ and $\mathbf{Y}^{\prime}$).
>
> The authors thank the reviewer for their suggestion. The definition now appears in our general comment and in the revised manuscript.
>
> > Theorem 4.1 and 6.1 claimed that two GNNs are "equivalent" but did not define "equivalent" mathematically. Therefore, I would suggest writing its formal definition.
>
> There are explicit mathematical proofs in the original manuscript for our notion of equivalence. In a nutshell, by equivalent we mean that they produce the same output.
>
> > (minor comments) Contribution (C2) claimed that "the WL algorithm is a limit [...] only when we use the all-one vectors as an input". Strictly speaking, this is not strictly true. It is true that the WL algorithm is upper bound when we use the all-one vector. However, it is not shown that the converse holds, i.e., there is the possibility that inputs are not all-one and the expressive power is lower than the WL algorithm.
>
> The reviewer is right. We will remove the word 'only'. However, the reviewer's comment also raises the concern of studying the expressive power of a function $\phi$ (the GNN in that case), by observing part of the function's domain, $\phi(\mathbf{1})$. As we mention in the general comment, this leads to ambiguous conclusions. That is one of the reasons why we studied random inputs that cover the whole domain of $\phi$.
>
> > Quality Contribution (C3) claimed that GNNs have a higher discriminative ability when "node features are initialized with random white noise." However, I think this is a slightly confusing expression. In my understanding, white random noise input means that we draw a sample from a Gaussian distribution and feed the sample to the GNN. However, the GNN in Figure 3(a) takes the expectation of the input variable. This operation is impossible from a single realization of a random variable . It requires the information of the random variable (e.g., distribution of ).
>
> See our general comment. By white random noise, we mean that the input $\mathbf{X}$ is a Gaussian random vector (not a single realization), which leads to an output that is also a random vector. Using this stochastic input model allows us to derive concrete conclusions on the expressivity of a GNN. By measuring the variance, we derive equivalent deterministic models. Note that our results apply to any distribution with zero mean and unit covariance, but we used Gaussian for simplicity. Also in practice, one can draw realizations from any distribution and measure the empirical variance or just use the equivalent models that we are proposing.

---

### Official Review · Reviewer_4qSA · 2022-10-25

**Confidence:** 3
**Correctness:** 3
**Technical Novelty And Significance:** 2
**Empirical Novelty And Significance:** 2
**Recommendation:** 5

**Clarity, Quality, Novelty And Reproducibility:**

The paper is clearly written. The novelty of the proposed model is limited but the analysis is good. Some assumptions are not well supported.

**Strength And Weaknesses:**

Strengths:

1. The motivation of analyzing GNNs from the spectral decomposition view is good.

2. The analysis of using random input features can help improve expressiveness is great.

Weakness:

1. The fundamental assumption that 1-WL only serves as the limit when using all-one vectors as input features is not clearly correct to me. Specifically, to prove that 1-WL is the limit of GNNs in the GIN paper, the only assumption is that the input feature space is countable. In other words, the input features do not have to be all-one vectors to achieve the proof. Hence, it is unclear to me why this fundament assumption hold in this paper.

2. The proposed new model uses features obtained from powers of matrix representations. However, the similar idea has been investigated in exiting work, such as SIGN [1]. Hence, the proposed model is not novel from this view.

[1] Rossi, Emanuele, et al. "Sign: Scalable inception graph neural networks." arXiv preprint arXiv:2004.11198 7 (2020): 15.


**Summary Of The Paper:**

This paper uses the spectral decomposition perspective to study the expressive power of GNNs. It argues that the 1-WL is not the real limit of GNN expressiveness. Instead, it only serves as the limit when we use all-one vectors as input features. Further, this paper proposes a new model, which uses features derived from powers of matrix representations and can be more expressive than regular GNNs.

**Summary Of The Review:**

Overall, I think this work has merits in terms of the motivation and the analysis. However, the technical assumption and contribution have not met the standard of ICLR.

---

> ### Author Response · Authors · 2022-11-18
> **Response to reviewer's comments**
>
> We would like to thank the reviewer for their detailed comments and valuable feedback.
>
> > The fundamental assumption that 1-WL only serves as the limit when using all-one vectors as input features is not clearly correct to me. Specifically, to prove that 1-WL is the limit of GNNs in the GIN paper, the only assumption is that the input feature space is countable. In other words, the input features do not have to be all-one vectors to achieve the proof. Hence, it is unclear to me why this fundament assumption hold in this paper.
>
> In the GIN paper, an implicit assumption is made that the input is the all-one vector or the degree vector, that is, $\mathbf{S}\mathbf{1}$, where $\mathbf{S}$ is the adjacency. This is due to the fact that 1-WL is initialized with the all-one vector or the degree vector (both admit the same limitations). If someone considers countable features, 1-WL is not the limit, as shown in our paper and other papers in the literature.
>
> > The proposed new model uses features obtained from powers of matrix representations. However, the similar idea has been investigated in exiting work, such as SIGN [1]. Hence, the proposed model is not novel from this view.
>
> See our general comment. In fact, most standard convolutional and message-passing GNNs are explicitly or implicitly using features obtained from powers of matrix representations. We never claimed that this is our contribution. Apart form using GNNs, there is no relation between our work and SIGN. Therefore, we would greatly appreciate the reviewer being more specific.

---

> > ### Comment · Reviewer_4qSA · 2022-11-27
> > **Follow-up discussions**
> >
> > Thanks for the response.
> >
> > > If someone considers countable features, 1-WL is not the limit, as shown in our paper and other papers in the literate.
> >
> > I did not get this. If the features are countable, 1-WL can represent them as one-hot vectors (the dimension equals to the cardinality of the feature space). Then, 1-WL is still the limit.

---

> > > ### Author Response · Authors · 2022-11-27
> > > **Response to follow-up discussions**
> > >
> > > We thank the reviewer for initiating this discussion.
> > >
> > > The first step in 1-WL assigns each vertex to a color corresponding to the vertex degree, i.e., $\mathbf{S}\mathbf{1}$ where $\mathbf{S}$ denotes the graph adjacency. We refer the reviewer to the original WL paper [1], which states: "Associate with every vertex of the graph the characteristic vector which has one component equal to the number of neighbors of this vertex". In our paper we explain that $\mathbf{S}\mathbf{1}$ admits the same limitations as $\mathbf{1}$.
> > >
> > > 1-WL cannot separate the graphs in Fig. 2 or any regular graphs, as in Fig. 1. Countable features as those in eq. 14 of the revised version (eq. 15 of the first version), overcome the limitations of 1-WL and distinguish between all graphs with different eigenvalues, as graphs in Figs. 1, 2.
> > >
> > > We think that the reviewer is referring to a comparison between GNNs with countable inputs and an algorithm that:
> > > 1) designs powerful countable features,
> > > 2) assigns an initial color to each vertex according to these features, and then
> > > 3) follows the 1-WL steps.
> > >
> > > This algorithm might be the limit of certain GNNs with countable features, however it is not the 1-WL algorithm. Furthermore, designing powerful input features requires the use of independent algorithms that can be more expressive than 1-WL. For example, consider the case where step 1, of the above algorithm, designs countable input features according to eq. 14 in the revised manuscript. These features are already distinct for non co-spectral graphs and step 2 and 3 are redundant.
> > >
> > > As mentioned in the general comment and the revised manuscript, one of our main contributions is that we show that GNNs can generate powerful representations from anonymous inputs and do need to rely on other algorithms to design powerful inputs. We prove that GNNs are more expressive than 1-WL and 2-WL algorithms, since they can produce countable features that 1- and 2-WL cannot produce and differentiate between graphs that 1-WL and 2-WL consider the same.
> > >
> > >
> > > [1] Weisfeiler, B. and Leman, A., 1968. The reduction of a graph to canonical form and the algebra which appears therein. NTI, Series, 2(9), pp.12-16.

---

### Official Review · Reviewer_GLDQ · 2022-10-25

**Confidence:** 3
**Clarity, Quality, Novelty And Reproducibility:** 1. **Clarity of proof of Theorem 2.2*…
**Correctness:** 3
**Technical Novelty And Significance:** 2
**Empirical Novelty And Significance:** Not applicable
**Recommendation:** 3

**Strength And Weaknesses:**

### 1. Strengths

- The paper provides expressivity analysis from a spectral perspective.
- The authors show that using random input features and a covariance operator yields a GNN model that is more expressive than 1-WL and show equivalence to a model that sidesteps the need to use random features. This result bridges betweenthe role of random features and substructure input features in increasing expressivity of GNNs.

### 2. Weaknesses

1. **Novelty**

   The paper suggests to augment the node features with the diagonal of adjacency powers, which are, the k-cycle couts (with repeatitions) for each node in the graph. This idea has appeared before e.g., in [1] that generally proposed to add information on subgraph isomorphism counts including cycles. Altough the proposed model was derived from a different thoretical perspective, I believe the novelty of the proposed model is quite limited.  The authos do not cite [1]. Can the authors provide a discussion upon relations of their work to [1]?

2. **Model complexity**

   Theorem 2.2 implies that in order to distinguish between two graphs the filter length should be at least $q$, the size of the set of all unique eigen values in the union of both graphs’ eigen values. This number grows with the number of nodes in the graph and could impose high complexity for large graphs.

3. **Too strong claims**

   The authors repeatedly state that the analysis of the expressive power of GNNs is limited to the case of an all 1 input vector. However, lots of works extended this result and the authors overview them in the related woprk section. I would suggest that the authors soften their claims dismissing other works that have already been done.

4. **Experimental Evaluation**

   Experimental evaluation is not convincing enough. The proposed method shows comperable performence to GIN where the improvement is insignificant. The results on the CSL dataset are not surprising, again since properties of the cycle counts features has already been analysed in [1]. Furthermore, [1] outperforms the current model across all social and biological datasets.



[1] Bouritsas, G.; Frasca, F.; Zafeiriou, S.; Bronstein, M. M. Improving Graph Neural Network Expressivity via Subgraph Isomorphism Counting. arXiv July 5, 2021.

###

**Summary Of The Paper:**

The paper provides a spectral analysis perspective on the expressivity of GNNs. The analysis is done under the assumption that graphs differ in at least one eigen value. The paper claims GNNs can achieve separation power greater than 1-WL by augmenting the node feature with appropriate choices and demonstrate the disadvantages of using an all-1 input vector. The authors suggest augmenting the input to the GNN with the diagonal of powers of the adjacency matrix to improve expressive power.

**Summary Of The Review:**

I think this work provides an interesting view on the analysis of GNN expressivity. However, the final proposed model lacks novelty and the experimental evaluation is not convincing enough.

I therefore think this paper is below the bar for ICLR but will consider raising my score upon addressing my concerns.

---

> ### Author Response · Authors · 2022-11-18
> **Response to reviewer's comments (2/2)**
>
> > Experimental evaluation is not convincing enough. The proposed method shows comperable performence to GIN where the improvement is insignificant. The results on the CSL dataset are not surprising, again since properties of the cycle counts features has already been analysed in [1]. Furthermore, [1] outperforms the current model across all social and biological datasets.
>
> The experiments aspire to provide insights on the effect of using anonymous-equivariant all-one inputs vs anonymous-equivariant random inputs (that translate to adjacency power diagonals) on the expressivity of GNNs. We study the plain-vanilla convolutional GNN model and the GIN model, which are standard architectures, that are falsely considered to be less expressive than the WL-test. We see statistical benefit in 3 datasets (CSL,REDDITBINARY,REDDITMULTI), whereas for the remaining datasets the performance is statistically the same. This is expected since the information loss due to the all-one vector is insignificant for the graph classification task in these datasets. The reviewer mentions that [1] has analyzed the properties of cycles, but this analysis is not indicative of the GNN representation capabilities, as defined in the general comment, but is due to the additional algorithms used in [1]. Therefore, further comparisons will not add to the value of the paper. After all, adding rich features, from any graph algorithm, is expected to benefit certain prediction tasks.

---

> > ### Comment · Reviewer_GLDQ · 2022-11-27
> > **Thank you**
> >
> > I thank the authors for their response.
> > See my reply on the general comment for a follow-up discussion.

---

> ### Author Response · Authors · 2022-11-18
> **Response to reviewer's comments (1/2)**
>
> We would like to thank the reviewer for their detailed comments and valuable feedback.
>
> > The paper claims GNNs can achieve separation power greater than 1-WL by augmenting the node feature with appropriate choices and demonstrate the disadvantages of using an all-1 input vector. The authors suggest augmenting the input to the GNN with the diagonal of powers of the adjacency matrix to improve expressive power
>
> From our general comment, it becomes clear that we do not just augment the diagonal of the adjacency powers to increase the expressive power of GNNs. We show that a GNN is powerful enough to produce the adjacency power diagonals and design an equivalent architecture where the inputs are indeed the adjacency power diagonals.
>
> > The paper suggests to augment the node features with the diagonal of adjacency powers, which are, the k-cycle counts (with repetitions) for each node in the graph. This idea has appeared before e.g., in [1] that generally proposed to add information on subgraph isomorphism counts including cycles. Altough the proposed model was derived from a different thoretical perspective, I believe the novelty of the proposed model is quite limited. The authors do not cite [1]. Can the authors provide a discussion upon relations of their work to [1]?
>
> We thank the reviewer for bringing [1] to our attention, which we cite in the revised version. [1] suggests augmenting the GNN layers with substructure information calculated from independent algorithms. The benefit compared to our work is that the count of various subgraphs can be explicitly computed (from other graph algorithms) and that one does not have to resort to training to retrieve this information. Note that exact subgraph counting is vital in certain tasks. Then, the training in [1] focuses more on how to combine subgraph counts rather than computing them, which leads to very interesting results, as reported in their paper.
>
>  We refer the reviewer to our general comment, which gives a good picture of the differences between the two works. To reiterate, contrary to [1], we do not resort to external algorithms to improve the expressive power of GNNs. We prove that GNNs can produce powerful and expressive outputs from naive inputs. Our results are completely based on the GNN architecture, compared to [1] that utilizes the expressive powers of subgraph counting algorithms and the results in [Xu et al., 2019]. Additional benefits of our work include our spectral analysis and explicit characterization of the class of graphs that a standard GNN can separate. We also provide conditions for general inputs. For example, Theorem 2.2 can be used to analyze the representational power of [1].
>
> > Theorem 2.2 implies that in order to distinguish between two graphs the filter length should be at least , the size of the set of all unique eigen values in the union of both graphs' eigen values. This number grows with the number of nodes in the graph and could impose high complexity for large graphs.
>
> In the worst-case scenario, there may be cases where the filter is in the order of the number of nodes. Note that this is expected since the diffusions performed in a GNN are local operations that try to capture global information. Having said that, in general this filter size is far from neccesary. Testament to this is the fact that $k=9$ has been used for the CSL graphs that have 41 nodes each. Also, for the graph in Fig. 2 that has 10 nodes, a filter with $k=5$ is employed, since 5-node cycles provide the distinctive information in that graph. Furthermore, Theorem 2.2 is an existence theorem and presents a particular instance that works for any graph. Instead, the Cayley-Hamilton theorem could have been used to prove the same result.
>
> > The authors repeatedly state that the analysis of the expressive power of GNNs is limited to the case of an all 1 input vector. However, lots of works extended this result and the authors overview them in the related woprk section. I would suggest that the authors soften their claims dismissing other works that have already been done.
>
> See our general comment. Our work argues that the analysis with all-one inputs or rich features is not sufficient to draw conclusions on the expressive power of GNN representations, as defined above. However, rich features are vital in increasing the performance of GNNs in separating between different graphs. We modify the discussion in the revised manuscript.

---

### Author Response · Authors · 2022-11-18
**General Comment**

We thank the reviewers for their helpful comments and constructive feedback.

We believe that the main point of our paper is being missed, so we provide a different way of presenting it.

A GNN can be defined as a function $\phi\left(\mathbf{X}; \mathcal{G}, \mathcal{H}\right)$, where $\mathcal{G}$ is the graph, $\mathbf{X}$ is the graph signal (input to the GNN), and $\mathcal{H}$ is the tensor of trainable parameters. We study the representation power of a GNN from the following perspective.

**Problem definition:**
Given a pair of different graphs $\mathcal{G}, \hat{\mathcal{G}}$ and anonymous inputs $\mathbf{X}, \hat{\mathbf{X}}$; is there a permutation matrix $\mathbf{\Pi}$ such that $\phi\left(\mathbf{X}; \mathcal{G}, \mathcal{H}\right)=\mathbf{\Pi}\phi\left(\hat{\mathbf{X}}; \hat{\mathcal{G}}, \mathcal{H}\right)$.

The above definition implies that a GNN is powerful, if it can **produce** nonisomorphic outputs from **anonymous inputs** for a large class of nonisomorpic graphs. By anonymous inputs, we mean inputs that are identity and structure agnostic, i.e., they cannot distinguish graphs or nodes of the graph before processing.

*Why anonymous?* Because if the inputs are discriminative prior to processing, we cannot derive concrete conclusions on the discriminative power of GNNs (no processing or little processing is enough). Note that we do not underestimate the importance of drawing powerful inputs. This is crucial for most tasks. However, we argue that powerful input features cannot deduce much about the expressive power of GNNs; they can only reveal if a GNN will maintain or dismiss valuable information, not if they can produce this information.

A natural choice for an anonymous input is the constant (all-one) vector. However, the constant vector is related to certain limitations (which we explain in the paper) and represents only a small subset of the domain of $\phi$. Thus, studying the codomain of $\phi$ with a constant input is necessary but not sufficient. As a result, we decide to examine the representational power of a GNN with white random input. White inputs are unonymous (they carry no information on the graphs) and they are also drawn from the whole domain of $\phi$.

We prove that GNNs are powerful, since they can produce discriminative outputs from white inputs, for at least all graphs that have different eigenvalues. In particular, we show that $\phi\left(\mathbf{X}; \mathcal{G}, \mathcal{H}\right)$, $\phi\left(\mathbf{X}; \hat{\mathcal{G}}, \mathcal{H}\right)$ belong to nonisomorphic distributions (the variance is different), even though the input $\mathbf{X}$ is the same (drawn from the same distribution). Furthermore, we show that the variance of $\phi\left(\mathbf{X}; \hat{\mathcal{G}}, \mathcal{H}\right)$ is related to the number of closed paths in which each node in $\hat{\mathcal{G}}$ is involved, and derive an equivalent model with deterministic features. Note that these features can be viewed as the output of the first GNN layer, i.e., they can be generated from a GNN.

Overall, the scope of the paper is to study the representational capabilities of GNNs and not to suggest rich inputs that improve the performance of GNNs with respect to graph isomorphism. Although this is a suttle difference, it is also important. We update the revised version of our manuscript with this discussion that provides a better understanding of the value of our work.

In the revised version, the analysis of the main paper is extended to graphs that have a different multiset of eigenvalues, i.e., they either differ in at least one eigenvalue or they have the same set of eigenvalues, but they differ in at least one eigenvalue multiplicity.

---

> ### Comment · Reviewer_GLDQ · 2022-11-27
> **follow up question**
>
> I thank the authors for their response and the effort put into clarifying their contribution.
>
> I am, however, still not sure I fully understand the general dismay the paper expresses about the analysis of the expressive power of GNNs. The expressive power of GNNs has been greatly explored beyond the all-1 input [1,2,3,4]. Both [3] and [4] show that GNNs with random features inputs (e.g., white noise) are **universal**; that is, they produce different outputs for non-isomorphic graphs. I therefore do not find the claims of this paper new and I think they are weaker in the sense that they only apply to non-isomorphic graphs with different eigenvalues.
>
> Can the authors comment on this point?
>
>
> [1] Murphy, et. al. [Relational Pooling for Graph Representations.](https://doi.org/10.48550/arXiv.1903.02541)
>
> [2] Sato, et. al. [Random Features Strengthen Graph Neural Networks.](https://doi.org/10.48550/arXiv.2002.03155)
>
> [3] Puny, et. al. [Global Attention Improves Graph Networks Generalization.](https://doi.org/10.48550/arXiv.2006.07846)
>
> [4] Abboud, et. al. [The Surprising Power of Graph Neural Networks with Random Node Initialization.](https://doi.org/10.48550/arXiv.2010.01179)

---

> > ### Author Response · Authors · 2022-11-28
> > **Response to the follow up question (2/2)**
> >
> > > I think they are weaker in the sense that they only apply to non-isomorphic graphs with different eigenvalues.
> >
> > We would like to remind the reviewer that our results do not only apply to non-isomorphic graphs with different eigenvalues. We study graphs with different eigenvalues in the main paper, since we will rarely encounter cospectral graphs in practice. However, Section H in the Appendix provides a rigorous analysis for graphs that share the same non-repeated eigenvalues. The only case we do not touch upon is graphs that have both the same multiset of eigenvalues and the eigenvalues are repeated. The likelihood of processing such graphs with GNNs is very small.
> >
> > Also, note that our analysis can easily show expressiveness for all graphs with probability one, at the cost of equivariance. As we mention earlier, this is not the point. The point it to prove expressiveness, while maintaining permutation equivariance. Our work proves that fact.

---

> > > ### Comment · Reviewer_GLDQ · 2022-12-06
> > > **Follow Up #2**
> > >
> > > Following our discussion, I think I gained a better understanding of this work’s contribution. Nevertheless, I think the current version of the paper does not properly convey the insights presented in it. A significant change re-framing the main takeaways and better placing this work among the existing literature is required. I am therefore leaning towards keeping my recommendation for rejection but encourage the authors to re-submit to another venue upon editing the paper.
> > >
> > >
> > >
> > > Below are the reasons for my decision and recommendations for improvement:
> > >
> > > - In the current version the motivation for exploring random inputs solely relies on Theorem 2.2. However, evidence for the expressive power gained by using random inputs has been shown in several works before. Noting these relations when suggesting the analysis of white noise would be fair.
> > >
> > > - The paper attributes a beyond 1-WL expressive power to GNNs with anonymous inputs, however, it is true for GNNs with an intractable variance non-linearity. That is, the GNN module in 3a cannot be exactly evaluated. Only the equivalence shown in equation (9) results in a computable module but that actually ends up with the need to assign non-anonymous inputs (diagonals of adjacency powers).
> > >
> > > - > but there is **no indication if expressivity is maintained in expectation**
> > >
> > >   I think this point is quit important and in fact this work shows that it is maintained in expectation and the authors should highlight it in the next version.

---

> > > > ### Author Response · Authors · 2022-12-07
> > > > **Response to Follow Up #2**
> > > >
> > > > We thank the reviewer for appreciating the contribution of our work. We also understand that the reviewer would prefer a different presentation of the main ideas. This is a fair opinion but an opinion nevertheless. We think it is not out of place to point out that papers should be evaluated on their scientific merit.
> > > >
> > > > With that in mind, we would like to clarify that some of the points that you make below are factually incorrect.
> > > >
> > > > > In the current version the motivation for exploring random inputs solely relies on Theorem 2.2.
> > > >
> > > > This is factually incorrect. Exploring the use of random inputs is the central purpose of our paper. Discussions of why random inputs are of interest appear in pretty much all sections of the paper. Among many other places, the role of random inputs is mentioned in the introduction; it is the **sole** subject matter of Section 4; and it is also one of the main points in Section 5.
> > > >
> > > > > However, evidence for the expressive power gained by using random inputs has been shown in several works before. Noting these relations when suggesting the analysis of white noise would be fair.
> > > >
> > > > Random inputs have been used in previous works as unique identifiers, thus losing the fundamental permutation equivariant property of GNNs. Being able to differentiate between the majority of non-isomorphic graphs, without being able to detect any isomorphic graphs, cannot lead to concrete conclusions about the expressive power of GNNs. These factual limitations are known and have been discussed in the literature. This is very different from what we claim in our paper which uses random inputs in a way that maintains permutation equivariance. We are including this discussion in the revised version.
> > > >
> > > > > The paper attributes a beyond 1-WL expressive power to GNNs with anonymous inputs, however, it is true for GNNs with an intractable variance non-linearity. That is, the GNN module in 3a cannot be exactly evaluated.
> > > >
> > > > This is factually incorrect. The **variance non-linearity** is **tractable**. The output has a **closed form representation** that makes evaluation of the GNN module in 3a elementary. It is the same as the module in 3b and equation (9).
> > > >
> > > > > Only the equivalence shown in equation (9) results in a computable module but that actually ends up with the need to assign non-anonymous inputs (diagonals of adjacency powers).
> > > >
> > > > This is factually incorrect. Equation (9) is the closed form expression for the variance of the filter output when the input is white. Equation (9) is not a GNN module with non-anonymous inputs. It is a **graph convolutional filter with no input**.

---

> > ### Author Response · Authors · 2022-11-28
> > **Response to the follow up question (1/2)**
> >
> > We truly value the reviewer's willingness to engage in a fruitful conversation. We also want to clarify that the reviewer is bringing up a set of issues that were not mentioned in the original review. We hope that this implies that the reviewer's original concerns are addressed in our previous response, and the discussion is now moving to the next level.
> >
> > > I am, however, still not sure I fully understand the general dismay the paper expresses about the analysis of the expressive power of GNNs.
> >
> > We do not express any emotion (dismay), but provide scientific facts. We particularly point out that there is an important point missing from the existing literature: GNNs are more expressive than the WL test. This does not require *expanded architectures* as in, e.g., [1, 3] or *expressive node features* as in, e.g., [1] (in reviewers previous comment), or *giving up permutation equivariance* as in, e.g., [2,3,4]. It just requires **carefully analyzing** the GNN with white inputs.
> >
> > > The expressive power of GNNs has been greatly explored beyond the all-1 input [1,2,3,4].
> >
> > Reference [1] proposes a high-order GNN and [3] introduces a novel attention mechanism. Both works study different architectures than standard GNNs, which is our topic of interest.
> > > Both [3] and [4] show that GNNs with random features inputs (e.g., white noise) are universal; that is, they produce different outputs for non-isomorphic graphs. I therefore do not find the claims of this paper new
> >
> > The reviewer is correct to say that the advantage of random features has been pointed out in [2] and [4]. However, these papers achieve enhanced function approximation **at the expense of permutation equivariance**. Permutation equivariance is a fundamental property in deep learning. Powerful GNNs produce equivariant representations for isomorphic graphs and distinct representation for non-isomorphic ones.
> >
> > 1) References [2] and [4] prove that random inputs produce expressive representations but **not equivariant**. Permutation equivariance in [2] and [4] is achieved in expectation, but there is **no indication if expressivity is maintained in expectation**. For example, our work shows that the expectation of a graph filter, although equivariant, carries zero information when the input is zero mean. It is also easy to show that when the input has constant mean, the expectation of the graph filter admits the limitations of the all-one input. Thus, there is an implicit trade-off between expressivity and equivariance. However, definitive results on the GNN performance require both properties.
> >
> >     Our work analyzes a GNN with random inputs and shows that the outputs are both **expressive** and **equivariant**. Specifically, the variance of the output is an equivariant, deterministic feature that **counts all the closed paths each node is involved in**, and is also **discriminative** for at least all graphs with different eigenvalues.
> >
> >     To further emphasize on the differences between our work and those in [2], [4] we will briefly explain why we maintain permutation equivariance and these works do not. The source of the problem is that [2], [4] employ randomness to draw realizations that provide unique identification of the nodes. Unique identifiers come with universality claims at the cost of permutation equivariance. Our work does not draw unique random samples, but treats white noise as an anonymous input to analyze the GNN output. This approach enables the derivation of an equivalent inputless GNN model (eq. 10 in the revised manuscript), which produces powerful equivariant representations.
> >
> > There is also another fundamental difference between our work and [2], [4]:
> >
> > 2) **The results in [2] and [4] are probabilistic:** As a result, they do not prove that a GNN can deterministically produce different outputs for non-isomorphic graphs; our work does. Given non-isomorphic graphs with different eigenvalues, we can always design a GNN that separates them.
> >
> > The above arguments also apply to the work in [3]. To be fair, we do not involve this work in the discussion, since the main contribution of [3] is on the architecture design, rather than the use of random features.
> >
> > 3) Our work offers **additional contributions**: In particular the **novel spectral approach**, the developed **general input theory**, the **analysis of the limitations of the all-one input**, and the **design of simple equivalent architectures**, are major innovations that are thoroughly described in our paper.

---

### Comment · Area_Chair_WRS7 · 2022-11-23
**Please discuss and respond**

Dear reviewers,

The authors have made an effort to write rebuttals to your reviews. Please read, think, and respond to the rebuttals. As we all know, most authors put a lot of effort into their work and the rebuttals, so the least we should do is acknowledge the responses.

Thank you.

AC

---

### Decision · Program_Chairs · 2023-01-20

**Decision:**

Reject

**Justification For Why Not Higher Score:**

All reviewers tended towards rejecting the paper. Two reviewers have a strong opinion about the paper having to get rejected. Especially one of these reviewers participated actively in discussions.

**Justification For Why Not Lower Score:**

N/A

**Metareview: Summary, Strengths And Weaknesses:**

The paper obtained low scores from the reviewers, even after a long discussion where the authors had time to argue their case. After reading the paper and reviews, I want to first encourage the authors to continue to pursue this line of work. Indeed, mainstream expressivity papers make assumptions that are strong and which are often not met in practice. Hence, the author's research is important. However, as rightfully criticized by the reviewers, the authors did not succeed in placing their work in the context of prior work that did not only consider all 1 vectors. There is also generally room for improvement in motivating the method and a lack of clarity in describing the method. Finally, the experiments are not very convincing in that there is only a minor, potentially insignificant improvement of their model compared to the state of the art. I don't believe, however, that this is a fundamental problem as long as the motivation, related work, and formal analysis of the method are improved.